# Inactivation of SARS-CoV-2 at acidic pH is driven by partial unfolding of spike
Irina Glas [1], Liv Zimmermann [2,3], Beiping Luo[4], Marie O. Pohl[1], Antoni G. Wrobel [5], Aline Schaub[6], Liviana K. Klein[4], Shannon C. David[6], Elisabeth Gaggioli[1], Nir Bluvshtein[4], Michael Huber [1], Athanasios Nenes[7,8], Ulrich K. Krieger [4], Thomas Peter[4], Tamar Kohn [6], Petr Chlanda [2,3] & Silke Stertz [1] ✉

SARS-CoV-2, the causative agent of COVID-19, is predominantly transmitted by respiratory aerosol and contaminated surfaces. Recent studies demonstrated that aerosols can become acidic, and acidification has been proposed as decontamination method. Here, we investigate how SARS-CoV-2 reacts to acidic pH and by which mechanism the virus is inactivated. We show that a pH below 3 is required to inactivate SARS-CoV-2 in a period of seconds to minutes. While we measured a 1000 to 10,000-fold drop in infectivity, virion structure remained intact under these conditions. Using super-resolution microscopy, we found that the attachment of virions to target cells is abrogated after acidic treatment, revealing spike protein (S) as the major inactivation target. Limited proteolysis of S combined with testing spike-specific antibodies for binding under low pH conditions revealed that exposure of SARS-CoV-2 to pH below 3 results in partial unfolding of S, thereby preventing binding of virions to target cells.

Severe acute respiratory syndrome coronavirus 2 (SARS-CoV-2) emerged in 2019 and has since caused a pandemic with over 7 million deaths worldwide according to the WHO[1]. SARS-CoV-2 belongs to the family of Coronaviridae, which are enveloped viruses with a positive sense RNA genome. The spike glycoprotein (S) is the main determinant facilitating viral entry. S is comprised of two subunits (S1 and S2) separated by a furin cleavage site that is either cleaved during viral egress by furin or furin-like proteases, during attachment to a host cell by transmembrane serine protease 2 (TMPRSS2) or during endosomal trafficking by Cathepsin L[2–7]. The S1 subunit includes the N-terminal domain (NTD) as well as the receptor binding domain (RBD), responsible for binding to the cellular receptor angiotensin-converting enzyme 2 (ACE2)[8,9]. After attachment, the S1 subunit dissociates together with ACE2 from S2[10,11]. Subsequently, S2 is cleaved at the S2' site by TMPRSS2 (fusion at the plasma membrane) or Cathepsin L (fusion in the endosome)[4,12]. This enables conformational changes that expose the fusion peptide and eventually lead to fusion, releasing the genome into the cytosol[4]. However, there is also evidence that S2' cleavage is not necessarily required for the transition of S2 into the postfusion conformation after ACE2 binding[12]. SARS-CoV-2 has four transmission modes: Direct contact, indirect contact through contaminated fomites, direct deposition of larger infectious respiratory particles (IRPs), and airborne transmission through inhalation of small IRPs[13,14]. Particularly the latter has been a challenging obstacle for transmission route tracking and disease containment. Therefore, in order to mitigate SARS-CoV-2 infection waves, other endemic respiratory viruses, as well as future zoonotic events, it is crucial to understand airborne transmission and develop new measures to control spread[15].

IRPs undergo dramatic physicochemical changes after aerosol emission from the respiratory tract until they reach chemical equilibrium with their gaseous environment. This includes changes in salt concentration, crystallization of salts, and potentially liquid-liquid phase separation or the formation of glassy shells[16–19]. Furthermore, volatile acidic and alkaline compounds can partition between the gas phase and IRPs, influencing their pH. In fact, outdoor aerosol particles can exhibit extreme acidity with pH below 1[20,21], drawing attention to pH as an interesting factor for virus stability in IRPs. Recent research has led to lively discussions about the precise

[1]Institute of Medical Virology, University of Zurich, Zurich, Switzerland. [2]Schaller Research Group, Department for Infectious Diseases, Virology, Heidelberg University, Heidelberg, Germany. [3]BioQuant - Research Center for Quantitative Analysis of Molecular and Cellular Systems, Heidelberg University, Heidelberg, Germany. [4]Institute for Atmospheric and Climate Science, ETH Zurich, Zurich, Switzerland. [5]Department of Biochemistry, University of Oxford, Oxford, UK. [6]Environmental Chemistry Laboratory, School of Architecture, Civil and Environmental Engineering, Swiss Federal Institute of Technology in Lausanne, Lausanne, Switzerland. [7]Laboratory of Atmospheric Processes and their Impacts, School of Architecture, Civil and Environmental Engineering, Swiss Federal Institute of Technology in Lausanne, Lausanne, Switzerland. [8]Institute of Chemical Engineering Sciences, Foundation for Research and Technology Hellas, Patras, Greece. ✉e-mail: stertz.silke@virology.uzh.ch

pH in respiratory aerosols, which remains uncertain and is a topic of ongoing research. Nevertheless, it has been shown that pH changes are variable and depend on several factors including aerosol matrix, time after emission from the respiratory tract, particle size, and air composition[22]. Microphysical model calculations and aerosol particle experiments suggest IRPs transition through an alkaline phase due to rapid loss of bicarbonate but then, depending on air composition and particle size, readily equilibrate at acidic pH due to uptake of ambient acids[22–25]. Model calculations also indicate that due to diffusive limitations, the time scale to transition to acidic pH decreases with particle size[23]. For example, it was predicted that the acidity of IRPs with a radius of 1 μm would reach pH 4 within about 2 min after exhalation in typical indoor air, while it takes hours to reach the same pH in particles larger than 20 μm[22,23]. Viral RNA, infectious virus and finally transmission in animal models has been detected for various respiratory particle size fractions, however, a majority of viral RNA was found in smaller particles[26–29]. Additionally, smaller IRPs remain airborne for longer, travel further and are captured by surgical masks to a lesser extent than larger particles[26].

Factors influencing IRP properties, such as IRP matrix, time after exhalation, and particle size, are not easy to influence, but gas phase compositions can be varied. The variation of relative humidity (RH) is the most prominent example of a gas phase property influencing stability of aerosolized viruses[30,31]. Recently, modifying air composition to shift aerosol pH (in either direction) has been proposed as a potential mechanism to mitigate transmission risk[15,23,24]. However, the sensitivity to acidic or alkaline pH differs between viruses. For example, influenza A viruses (IAVs) are readily inactivated at pH below 5.5–6, while coronaviruses are more stable at acidic pH: in a previous study, we observed inactivation of SARS-CoV-2 only at pH below 3. Interestingly, human coronavirus 229E (HCoV-229E) remained stable for hours even at pH 2[23]. These virus-specific differences suggest distinct inactivation mechanisms. While the inactivation mechanism of IAV at low pH is understood to be caused by the hemagglutinin glycoprotein (HA) irreversibly switching to its post-fusion conformation[32], the inactivation mechanism of SARS-CoV-2 at acidic pH has been unknown. To date, research concerning S structure has focused on pH values down to 4[33,34], representing physiological pH in endosomal compartments of cells, at which SARS-CoV-2 remains infectious[23]. In this study, we investigate the inactivation mechanism of SARS-CoV-2 at pH that is lower than intracellular values but can be encountered by viruses in small IRPs when certain gas phase composition requirements are met. We show that partial unfolding of S is the primary driver of SARS-CoV-2 inactivation at acidic pH disrupting the viral life cycle at the attachment stage. However, HCoV-229E S is more stable and virions remain infectious for hours at pH 2.0. These findings contribute to our understanding of coronavirus stability in IRPs.

## Results

### Acidic pH below 3 leads to fast inactivation of SARS-CoV-2, but not HCoV-229E

Respiratory aerosol pH is a crucial factor concerning aerosol stability of viruses with varying impacts on different viruses[23]. The final pH and time to reach acidic pH is, amongst other factors, such as the matrix composition, determined by the gas phase composition indoors with nitric acid (HNO$_3$) being a major player. Biophysical modeling with the respiratory aerosol model ResAM shows that the pH of an IRP with an initial radius of 1 μm drops drastically to values slightly below 4 within seconds to a few minutes after emission from the respiratory tract in typical indoor air (Fig. S1A)[23]. In contrast to IAVs, coronaviruses have been shown to be stable at pH above 3, suggesting that acidic pH is not a critical factor for coronaviruses in a typical indoor transmission scenario[23]. Increasing indoor HNO$_3$ concentrations to 10 or even 50 ppbv, well below the CDC-established regulatory 8-h exposure limit of 2000 ppbv, would be enough to further lower the pH of IRPs to ~2 within seconds to minutes, as determined by ResAM (Fig. S1A, B)[35]. To investigate the inactivation mechanism of coronaviruses at pH encountered in IRPs suspended in

acidified air, we exposed the early SARS-CoV-2 isolate BetaCoV/Germany/BavPat1/2020 (BavPat1)[36] and HCoV-229E encoding a *Renilla* luciferase reporter (HCoV-229E-Ren)[37] to pH below 3 in bulk solution. In brief, virus samples were exposed to an acidic environment, incubated at RT for various periods, and then neutralized as depicted in Fig. 1A. Neutralization is an essential step in our setup, since IRPs also return to neutral pH when they are re-inhaled by a susceptible person. We found that SARS-CoV-2 is stable at pH 7.1, while substantial inactivation was observed at pH 2.8, with more than a 1000-fold drop in titer after 10 min of exposure. At pH 2.2, we observed fast inactivation with a nearly complete inactivation already after 10 s, as shown in Fig. 1B. We then performed single-cycle infection experiments with pH-treated BavPat1 samples on Vero-E6 cells overexpressing ACE2 and TMPRSS2 (VAT cells), and tested for the production of nucleocapsid (N) protein by immunofluorescent (IF) staining 7 h post-infection (hpi). In line with the plaque assay titrations, no viral protein production was observed when cells were infected with SARS-CoV-2 treated at either pH 2.8 for 10 min or pH 2.2 for 10 s (Fig. 1C), suggesting that early stages of the infection were abrogated. Remarkably, HCoV-229E-Ren showed high stability at acidic pH with no detectable inactivation after 1 h of exposure to pH 2.0 as shown in Fig. 1D. The *Renilla* gene did not influence these results, as we observed the same results with the HCoV-229E wild type strain (Fig. S1C). To test if this inactivation kinetic of SARS-CoV-2 BavPat1 is representative of currently circulating variants, we studied the JN.1 Omicron isolate SARS-CoV-2/human/Switzerland/ZH-UZH-IMV15/2024 (JN.1). The stability of JN.1 at pH 7.1 and 2.1 was comparable to BavPat1 (Fig. S1D). However, we observed a faster inactivation of JN.1 at pH 2.7 with an ~100-fold higher drop in infectivity as compared to BavPat1 after 1 min of exposure (Fig. S1E). Nevertheless, both strains are stable at pH 4.3 over the measured time course, which is representative of the predicted equilibration pH of IRPs in typical indoor air, and are therefore considerably more stable than IAV (Fig. S1F). Although inactivation times between SARS-CoV-2 variants differ at pH 2.7, overall, we conclude that BavPat1 is a suitable SARS-CoV-2 representative. As virus stock preparation and titer varied between our SARS-CoV-2 and HCoV-229E samples, we tested the impact of stock concentration methods on pH stability, as well as the impact of virus titer during the pH exposure reaction. We did not observe any change in HCoV-229E-Ren stability within 10 min exposure to pH 2.0 when the input titer was 10-fold lower or when stocks concentrated by ultracentrifugation or Amicon filtration were tested (Fig. S1G,H). Even after 1 h of exposure to pH 2.0 we could not observe a difference between methods used to concentrate HCoV-229E-Ren virus stocks or between different reaction titers used (Fig. S1I,J). These data highlight that HCoV-229E is indeed substantially more pH stable than SARS-CoV-2.

### Viral membrane integrity of SARS-CoV-2 is preserved after acid treatment

First, we used cryo-electron tomography (cryo-ET) to explore if there are any major differences in the virion morphology of SARS-CoV-2 and HCoV-229E concerning shape and S number, potentially explaining the difference in stability. For cryo-ET and following integrity analyses, we used SARS-CoV-2 BavPat1 and HCoV-229E-Ren as representatives for SARS-CoV-2 and HCoV-229E. Exemplary cryo-ET slices are shown in Fig. 2A. SARS-CoV-2 and HCoV-229E virions have a similar average diameter of 103 nm and 104 nm, respectively (Fig. 2B). Moreover, the number of S proteins per virion with an average of 19 S per SARS-CoV-2 virion and 23 per HCoV-229E virion do not show a significant difference (Fig. 2C). However, HCoV-229E virions were more elongated compared to the more spherical SARS-CoV-2 virions (Fig. 2D). Furthermore, 5.3% of SARS-CoV-2 S proteins were in the post-fusion conformation compared to 1.1% of HCoV-229E S as depicted in Fig. 2E. Although the number of post-fusion S is higher for SARS-CoV-2 particles, the vast majority of S is in the pre-fusion conformation and therefore considered functional. Thus, our cryo-ET analysis does not reveal any major differences in virus morphology of SARS-CoV-2 and HCoV-229E.

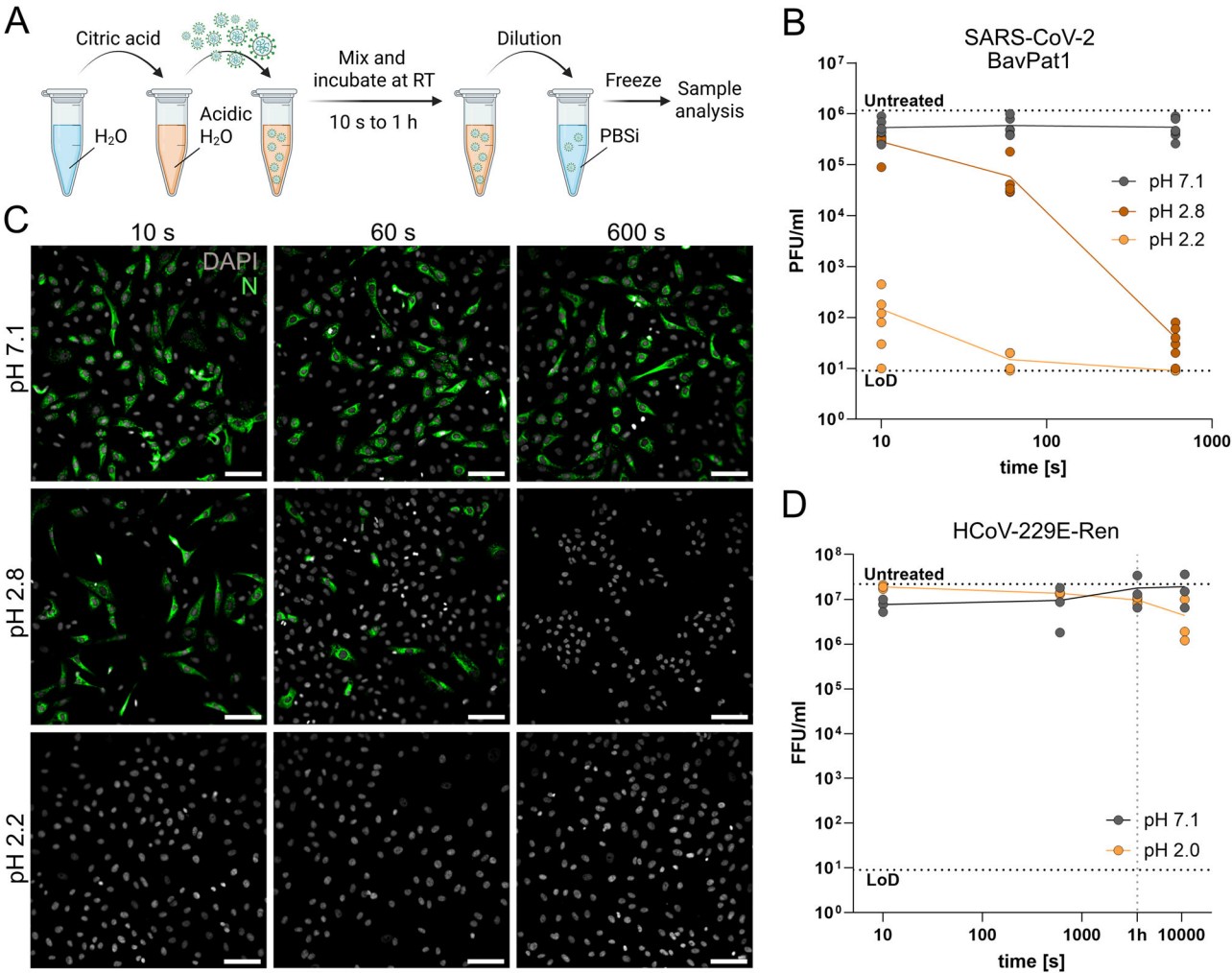

**Fig. 1 | SARS-CoV-2 BavPat1, but not HCoV-229E-Ren, is efficiently inactivated at a pH of 2.2. A** Scheme showing the sample preparation in bulk solutions. Virus is spiked into an acidic environment and incubated at RT. At each time point, a sample is taken from the reaction and neutralized by a 1:100 dilution in PBSi (additionally buffered at pH 7). Created in BioRender. Glas, I. (2025) https://BioRender.com/o9x4cqe. **B** Inactivation curves in PFU/ml of SARS-CoV-2 BavPat1 over time at pH 7.1, 2.8, and 2.2 determined by plaque assay from three independent experiments done in duplicates. Samples were generated as shown in (**A**). Dotted lines indicate titer measured in the untreated control and the limit of detection (LoD). **C** IF images of VAT cells infected with neutralized SARS-CoV-2 BavPat1 samples treated for 10 s, 1 min (60 s) or 10 min (600 s) at pH 7.1, 2.8 or 2.2. Infected VAT cells were fixed at 7 hpi and subsequently stained for SARS-CoV-2 N (green) and with Dapi (gray). Shown are representative images from two independent experiments done in duplicates. Scale bar corresponds to 100 μm. **D** Inactivation curves in Foci forming units (FFU)/ml of HCoV-229E-Ren over time at pH 7.1 and 2.0 measured by foci assay from three independent experiments. Dotted lines indicate titer measured in the untreated control and the LoD as well as the 1 h time point.

To investigate whether exposure to acidic pH causes major damages to virion structure, we utilized an RNase digestion assay as described by David et al.[32], in which virions are exposed to RNases, testing their capacity to protect the genome. To validate our assay set-up, we first tested viral RNA levels after RNase treatment of free viral RNA, SARS-CoV-2 virus samples as well as ethanol-treated SARS-CoV-2 virus. Free RNA was efficiently digested with an average reduction around 10,000-fold, while SARS-CoV-2 stock samples had no detectable losses in genomic copy (GC) numbers (Fig. 2F). If SARS-CoV-2 stock samples were treated with 80% ethanol disrupting the membrane, RNases could access the viral genome, and the amount of extracted RNA was reduced by an average of around 2500-fold (Fig. 2F). When applying the assay to low pH-treated samples of SARS-CoV-2 we did not observe striking changes in the RNA levels (Figs. 2G and S2A). Infectious titer and GC reduction did not correlate, with observed maximal reductions of 100,000-fold in PFU and 14-fold in GC (Fig. 2H). Therefore, we conclude that SARS-CoV-2 inactivation by low pH in Fig. 1B, C is not caused by envelope disruption. To assess the effect of acidic pH on virion structure, we performed cryo-ET on SARS-CoV-2 and HCoV-229E

samples after treatment at neutral or acidic pH with subsequent neutralization and UV-inactivation. To obtain higher virus concentrations for cryo-ET, we increased the amount of concentrated SARS-CoV-2 stock added into the pH reaction, hence also increasing protein concentrations in the inactivation reaction. Besides an increased buffering capacity impeding acidification, we observed a slower inactivation kinetic (Fig. S2B), most likely caused by the elevated protein content exerting protective effects[23,31,38]. We therefore increased the incubation time to a period of 1 h, at which almost complete inactivation was achieved, similar to pH 2.8 after 10 min or pH 2.2 after 10 s in a low-protein context (Fig. 1B). We additionally performed the RNase digestion assay on these samples, shown in Fig. 2I, and again found no relation between PFU reduction and GC reduction after RNase treatment (Fig. S2C).

Cryo-ET revealed that SARS-CoV-2 virions treated at pH 2.8 have an intact membrane and S on their surface. By eye, the amount of S per particle appeared to be decreased compared to virions treated at pH 7.1 (Fig. 2J). However, virion concentration of SARS-CoV-2 after neutralization was too low to acquire a sufficient number of tomograms (N = 3) for quantifications,

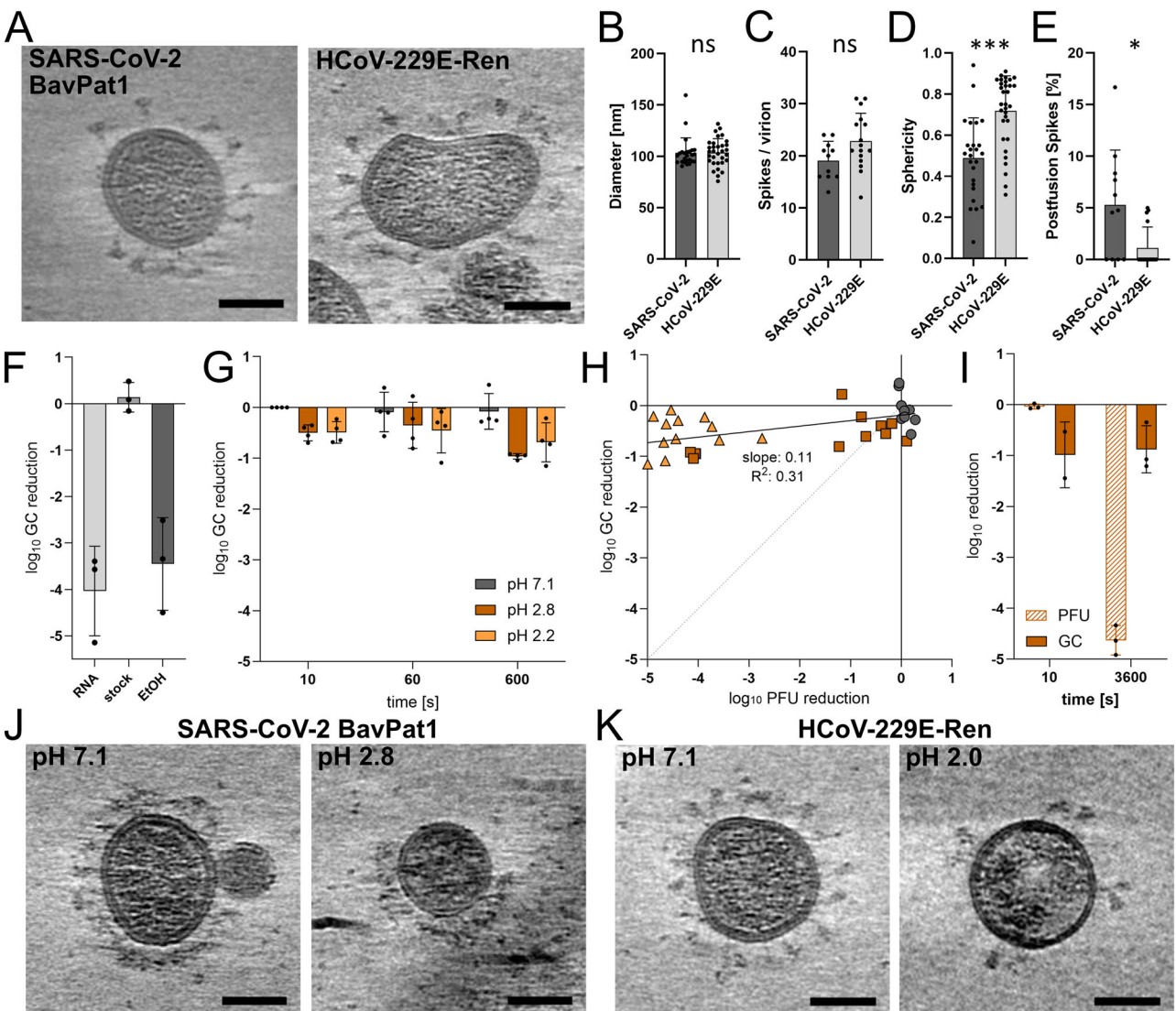

**Fig. 2 | Virion characterization and integrity of pH treated SARS-CoV-2 BavPat1 and HCoV-229E-Ren samples. A** Slices of tomograms showing virions from cell culture supernatants of SARS-CoV-2 BavPat1 and HCoV-229E-Ren infected cells. The scale bar corresponds to 50 nm. Quantification of untreated SARS-CoV-2 BavPat1 and HCoV-229E-Ren tomograms from n = 25 and n = 32 virions, respectively: Virion diameter in nm (**B**), number of S per virion (**C**), sphericity (**D**), and percentage of S in the post-fusion conformation (**E**). Tomograms were derived from a single SARS-CoV-2 BavPat1 or HCoV-229E-Ren sample. Statistical tests were performed using the unpaired, nonparametric Mann–Whitney test (two-tailed) in GraphPad Prism. Data was considered non-significant (ns) if p values were 0.05 or above and significant for p values below 0.05 (p < 0.05 (\*), p < 0.005 (\*\*), p < 0.0005 (\*\*\*)). **F** Control panel introducing the RNA digestion assay. Intact SARS-CoV-2 BavPat1 virions are able to protect their genetic material from nucleases, while free or accessible RNA is digested during RNase treatment. Included are a free RNA control (RNA), a virus stock sample (stock) and a virus stock sample lysed with 80% ethanol (EtOH). GC numbers were determined by qPCR on the E segment and the GC loss

calculated and $\log_{10}$ transformed. Shown are average values from n = 3 independent experiments. **G** Calculated $\log_{10}$ loss of GC after RNA digestion normalized to the pH 7.1 10 s sample. SARS-CoV-2 BavPat1 samples were generated as shown in Fig. 1A, subsequently exposed to RNases, and GC numbers were determined by qPCR. Shown are average values from n = 4 replicates derived from three independent experiments. **H** Data in (**G**) correlated with the corresponding $\log_{10}$ of the loss in PFU/ml of each sample calculated from the data shown in Fig. 1B. Linear regression was performed using GraphPad Prism (n = 36). The coefficient of determination ($R^2$) is 0.31. The slope is 0.11 (95% CI: [0.05, 0.17]), with a p value of 0.0004. Dotted line represents a perfect correlation. **I** As described in (**G**), $\log_{10}$ loss in GC numbers and PFU were calculated for SARS-CoV-2 BavPat1 samples with increased reaction titer and protein content during the pH inactivation at pH 2.8. Data was derived from n = 3 independent replicates. Slices of tomograms showing SARS-CoV-2 BavPat1 virions (**J**) or HCoV-229E-Ren virions (**K**) treated at pH 7.1, pH 2.8, or pH 2.0. The scale bar corresponds to 50 nm. In (**B–G, I**) data are means with the error bars representing the standard deviation (SD).

therefore we were unable to draw conclusions about a potential loss of S from the virions surface. Notably, SARS-CoV-2 virions treated at pH 2.8 showed signs of a disrupted inner structure, presumably corresponding to structurally altered viral ribonucleoprotein complexes (vRNPs) (Fig. 2J). A similar phenotype was observed for HCoV-229E virions treated at pH 2.0 with minor effects on infectivity (Figs. 1D and 2K), suggesting that the disruption is not detrimental to the entry process. Overall, we conclude that virion membrane of SARS-CoV-2 remains intact upon pH 2 treatment.

## SARS-CoV-2 virions inactivated by acidic pH lack binding ability to target cells

After confirming that virion membrane integrity is preserved, we examined the effects of acidic pH treatment on SARS-CoV-2 entry, starting with attachment to target cells, again choosing SARS-CoV-2 BavPat1 as a representative strain for SARS-CoV-2. To obtain a sufficient titer for the virus binding assay, we used the same approach as for cryo-ET and increased the amount of SARS-CoV-2 stock in the pH reaction and let the

**Fig. 3 | SARS-CoV-2 BavPat1 virions lose cell binding ability after low pH treatment.** VAT cells were incubated with neutralized SARS-CoV-2 BavPat1 samples (treated with pH 7.1 or pH 2.8 beforehand) for 1.5 h on ice. Subsequently, VAT cells with bound virus were fixed, permeabilized, stained for N (cyan) and with CellBrite 680, a membrane dye (brown), and imaged by super resolution microscopy. Shown are maximal z-projections of representative cells as well as extended y-z-sections and x-z sections as indicated with yellow lines. The red boxes show magnified images of the x-z-section. Scale bars correspond to 10 μm. Four to eight z-stacks were acquired per condition in each of three independent experiments.

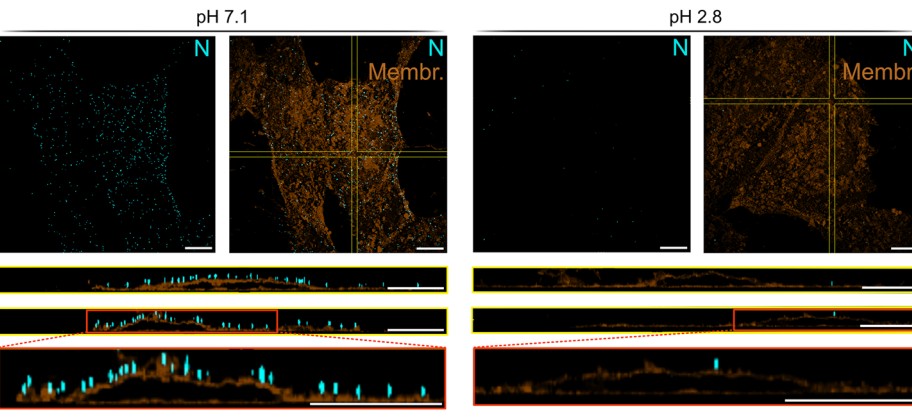

reaction proceed for 1 h to achieve almost complete inactivation at pH 2.8 (Fig. S2B). Neutralized SARS-CoV-2 samples treated with pH 7.1 or pH 2.8 were added to VAT cells and incubated for 1.5 h on ice to allow for virus attachment but prevent internalization[39]. We then imaged attached SARS-CoV-2 virions at the cell membrane by super-resolution microscopy using a membrane staining and an N antibody to detect SARS-CoV-2 virions. We found a pronounced loss of virions located at the plasma membrane of cells incubated with samples treated at acidic pH compared to samples treated at neutral pH (Fig. 3). This indicates that binding of SARS-CoV-2 virions to target cells was almost completely absent due to the inactivation at acidic pH. To further assess the link between virus inactivation and inability to attach to target cells, we performed the assay with samples treated at pH 7.1 or pH 2.8 for just 10 s. Both are conditions where no inactivation had been detected (Fig. S2B). In addition, we included an untreated SARS-CoV-2 and a mock control. As expected, no bound virions were observed in the mock control, while the untreated sample and the samples treated for 10 s showed numerous SARS-CoV-2 virions bound to the cell surface independent of pH treatment (Fig. S3A–D). Of note, we detected similar GC reductions in our RNase digestion assay for the corresponding samples treated at pH 2.8 for either 10 s or 1 h shown in Fig. 2I. Samples treated at pH 2.8 for 10 s showed an average GC reduction of 9.8-fold, slightly higher than the 7.4-fold observed for samples treated at pH 2.8 for 1 h (Fig. 2I), emphasizing that disrupted virion membrane integrity is not responsible for the observed reduction in bound virions in Fig. 3.

## Acidic pH below 3 induces irreversible conformational changes in S

After having mapped the defect of low pH inactivated SARS-CoV-2 to attachment, we aimed to study the conformational integrity of S in response to acidic pH treatment. We utilized limited proteolysis (LiP) as an approach to track conformational changes in recombinant S trimers treated at pH 2.0 in relation to the pH 7.0 control. In brief, recombinant S was treated with neutral or acidic pH, neutralized, concentrated and then exposed to proteinase K for 5 min. Subsequently, samples were denatured and fully digested with trypsin to allow bottom-up liquid chromatography coupled to mass spectrometry (LC-MS) analysis. We then looked for peptide abundance changes of pH 2.0 treated S compared to pH 7.0 treated S and identified six semi-tryptic peptides with significant changes (Fig. 4A). Four of those surpassed the $\log_2$ fold change threshold of 1. These four peptides belong to three different regions in the S2 domain (Fig. 4B). We noticed that the proteinase K cleavage sites of these peptides are buried inside the S trimer (Fig. 4B). This suggests inaccessibility of these cleavage sites in the native conformation (at pH 7.0) and in consequence increased proteinase K exposure due to conformational changes or trimer dissociation caused by the pH 2.0 treatment. The two remaining peptides that did not cross our $\log_2$ fold change threshold but reached significance are found in the NTD of the S1 subunit (Fig. 4B) and in the flexible region at the C-terminus of the S2 subunit, suggesting that other

regions might be affected as well. A list of all identified peptides can be found in Supplementary Data 1.

The S coverage in the LiP experiment was ~35% with most peptides located in the S2 region raising questions about undetected changes in the S1 subunit, and especially the RBD, that could explain the disruption in attachment (Fig. 3). Furthermore, the experiment was done with a stabilized trimer including furin cleavage site mutations as well as mutations favoring the pre-fusion conformation of S, which could potentially dampen pH effects. We therefore utilized various S-specific antibodies to study S conformation on virions. In brief, virions were bound to cover slips and exposed to different pH values for 1 min before neutralization, fixation, and IF staining. We tested two different neutralizing RBD antibodies recognizing SARS-CoV-2 BavPat1 S in a native trimeric conformation, as found on virions in IF and one RBD antibody recognizing the denatured, linear form of S, as found in western blots (WB). Further, we used two NTD antibodies and one S2 antibody with a preference for the denatured state of SARS-CoV-2 BavPat1 S. As a control we used an N antibody that binds to SARS-CoV-2 independent of pH treatment (Fig. S4A–C). All antibodies were tested in IF and WB to confirm binding preferences (Figs. 5A–C, S4D, E, and S5). Antibodies detecting S in WB were considered as antibodies with a preference for linear epitopes. We observed decreased binding of both neutralizing RBD antibodies to virions treated at pH 2.0 (Figs. 5A and S4D). However, all antibodies with a preference for the denatured form of S, including the RBD antibody, showed an increased signal in IF indicating better linear epitope accessibility due to the pH 2.0 treatment (Figs. 5A–C and S4E). We quantified the signal intensities by calculating the total intensity per image of all pixels positive for the N signal. For each image, the total intensity of the different S antibodies was normalized to the corresponding N signal intensity. Concerning neutralizing RBD antibodies, the RBD-to-N signal ratio declined with decreasing pH, with the strongest reduction observed after pH 2.0 treatment (Figs. 5D and S4F). To assess signal co-localization independently of signal strength, we additionally calculated the Manders correlation coefficient (M1) between the RBD signal and the N signal. The Manders coefficient reflects the fraction of the N signal overlapping with any detectable RBD signal, regardless of intensity correlation. In line with the intensity reduction, Manders' correlation values for neutralizing RBD antibodies also decreased at lower pH, indicating a partial loss of detectable RBD signal on virions (Figs. 5E and S4G). In contrast, the S2 antibody showed high Manders correlations (averaging 0.81) at pH 7.0 increasing to 0.96 in samples treated at pH 2.0 (Fig. 5G). Similarly, the RBD antibody #63847 with a binding preference to denatured S showed an increase in both intensity ratio and Manders correlation (Fig. 5H, I). The most pronounced effect was observed for the NTD antibody MA5-36247: While the signal was almost undetectable at pH 7.0, we revealed a strong increase in signal for SARS-CoV-2 BavPat1 virions treated at pH 2.0 with a fold change of ~12 in the intensity ratio and 3 in the Manders correlation (Fig. 5J, K). A similar effect was observed for the second NTD antibody #56996 (Fig. S4H, I). The raw intensity sums of the acquired images are provided in Fig. S4A–C, J, K.

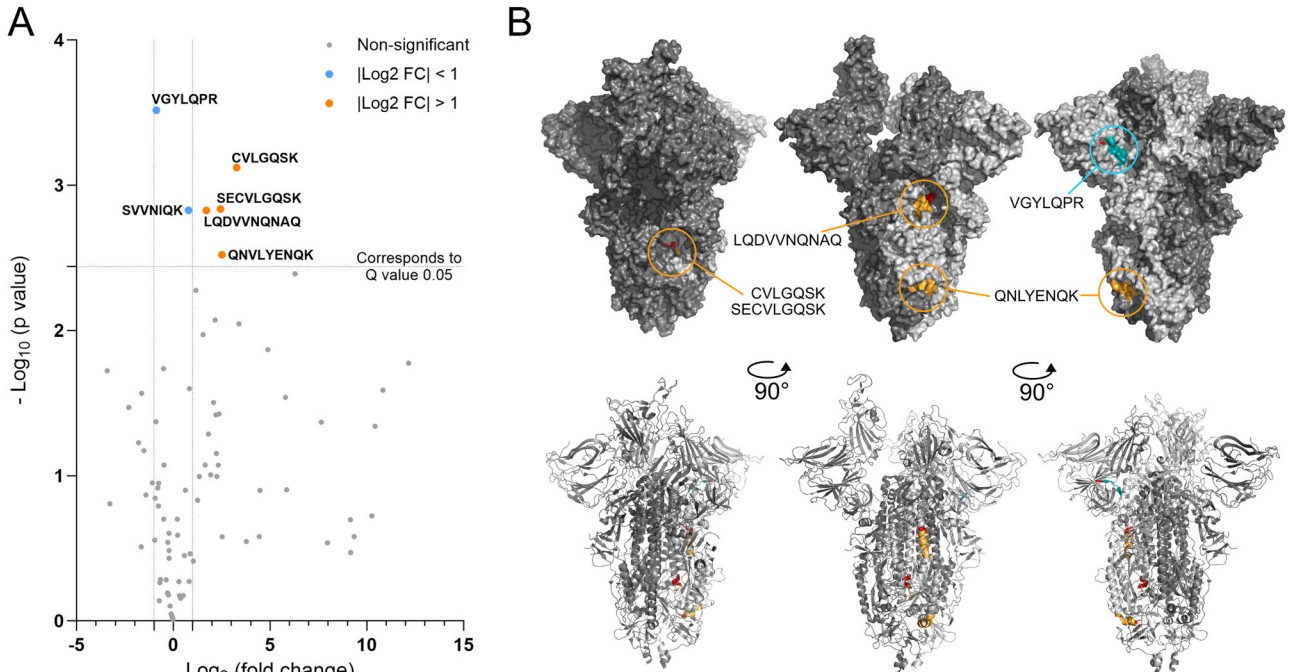

**Fig. 4 | Peptide candidates determined by LiP LC-MS after pH treatment of recombinant S. A** Volcano plot showing the $Log_2$ fold change of all S peptides derived from the pH 2.0 sample compared to the pH 7.0 sample. The negative $log_{10}$ p values are shown on the Y-axis. Peptides with a higher negative $log_{10}$ p value than 2.439 (corresponding to a Q value of 0.05) were considered significant. Dotted lines indicate $log_2$ fold changes of $-1$ and 1 as well as the significance threshold. Peptides with significant changes above a $log_2$ fold change of 1 were colored orange, while peptides with significant changes below 1 were colored blue. Data was derived from n = 4 technical replicates per condition. Raw data are available via ProteomeXchange

with identifier PXD064612. **B** S trimer structure (PDB ID: 6ZGG) from different angles showing the location of significant peptides. The upper row shows the surface view, while the lower row shows the cartoon view. Peptides with significant changes were visualized on a single monomer (light gray). Significant peptides above a $log_2$ fold change of 1 were colored orange, while peptides with significant changes below 1 were colored blue. Proteinase K cleavage sites were colored red for each peptide. Of note, the peptide "SVVNIQK" is in an unresolved region of S2 and therefore not depicted.

The observed decrease in signal from the neutralizing RBD antibodies detected by IF could be explained by two potential mechanisms: Either acidic pH causes shedding of the S1 subunit, or the low pH leads to conformational changes reducing the accessibility or disruption of the RBD. Both mechanisms could potentially result in decreased attachment of SARS-CoV-2 BavPat1 to the host cell after treatment with acidic pH (Fig. 3). To address this, we determined the ratio of cleaved S2 to uncleaved S0 in our virus preparations by WB and found that all SARS-CoV-2 BavPat1 preparations contained a mix of uncleaved and cleaved S with S2/S0 ratios ranging between 1.0 and 2.2 (Fig. 5L, M). Considering that S1 shedding could only occur for S in its cleaved form together with the increased signal of RBD and NTD antibodies (preferring linear epitopes) to pH 2.0 treated SARS-CoV-2 BavPat1 virions detected by IF (Figs. 5 and S4), we conclude that the reduction in RBD signal is unlikely caused by S1 shedding. In Figs. 5N and S4L we visualized the epitopes of all antibodies used, reported the structural preference for epitopes (linear or native) as determined by WB and IF, and summarized if acidic pH caused a signal increase or decrease in our IF assay. Overall, all antibodies binding linear epitopes showed a signal increase in our IF assay when virions had encountered acidic pH, while antibodies binding natively folded S showed a signal decrease. This is consistent with S unfolding events and denaturation disrupting native epitopes, while exposing linear epitopes. Most strikingly, we were able to detect unfolding in the RBD domain responsible for virion attachment, which was not detectable with the LiP approach.

**S structure is impaired in the SARS-CoV-2 variant JN.1 upon low pH treatment but to a lesser extent in HCoV-229E-Ren**
To assess if the results obtained for SARS-CoV-2 BavPat1 are reflective of other variants we repeated the antibody assay shown in Fig. 5 with the SARS-CoV-2 Omicron variant JN.1. For this purpose, we used a

neutralizing RBD antibody specific to Omicron variants with a preference for native epitopes as well as the SARS-CoV-2 S2 antibody with a preference for linear epitopes. Again, the binding preference of these antibodies was confirmed beforehand in IF and WB (Figs. S6A, B and S7A). Importantly, we found decreased binding of the RBD antibody to SARS-CoV-2 JN.1 virions treated at pH 2.0 compared to SARS-CoV-2 JN.1 virions treated at pH 7.0. This was reflected in a decreased intensity ratio and Manders correlation (Figs. 6A, B and S6A) suggesting native epitope disruption and therefore RBD unfolding. As Omicron variants such as JN.1 preferentially adopt the closed S conformation[40], these results show that the closed trimer conformation does not protect from conformational changes in the RBD induced by low pH. Although not significant, the SARS-CoV-2 S2 antibody showed a signal increase for SARS-CoV-2 JN.1 virions that encountered acidic pH (Figs. 6C, D and S6B), indicating that unfolding might also affect the S2 domain as observed for SARS-CoV-2 BavPat1. Finally, we employed the same assay to test the pH 2.0 insensitive virus HCoV-229E-Ren. We used two HCoV-229E S antibodies, one binding the S1 subunit and one binding to S2 (Fig. 6E–H). Both antibodies were determined to have a linear epitope preference (Figs. S6C, D and S7B). Since HCoV-229E is substantially more stable at pH 2.0 than SARS-CoV-2 we expected to detect less unfolding of S and indeed we did not observe changes in the intensity ratio of the S2 antibody (Fig. 6G) and in Manders correlation for both antibodies (Fig. 6F, H). However, we found a significant increase in the signal intensity ratio for the HCoV-229E S1 antibody (Fig. 6E), suggesting that there are slight changes in S structure, although the infectivity is retained. The raw intensity sums of all acquired images are provided in Fig. S6E–H. To set measurements across different viruses in relation to each other, we calculated the $log_2$ fold change of the intensity ratios measured for all antibodies, shown in Fig. 6I. Interestingly, the only two antibodies that were below the $log_2$ fold change threshold of 1 were the HCoV-229 antibodies, highlighting

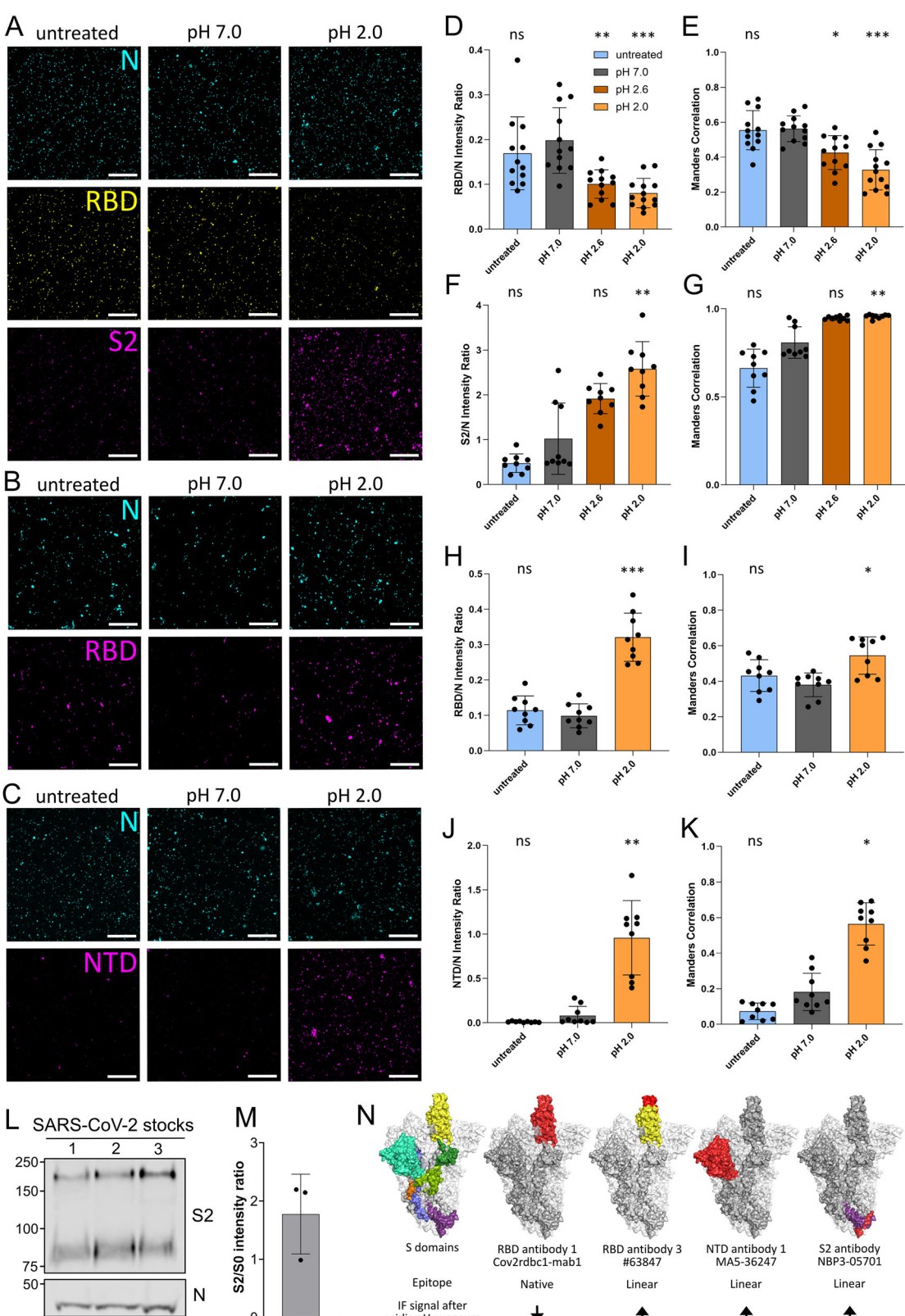

that HCoV-229 S is more stable compared to SARS-CoV-2 S at pH 2.0. The strongest effects were observed for the two NTD antibodies preferentially binding to the denatured form of SARS-CoV-2 BavPat1 S and neutralizing RBD antibodies binding either SARS-CoV-2 BavPat1 or JN.1, emphasizing that pH 2.0 efficiently induces unfolding events in the S1 subunit of SARS-CoV-2.

In sum, our results show that SARS-CoV-2 virion membrane integrity is not impaired by acidic pH even in extreme conditions like pH 2.2. However, SARS-CoV-2 virions lose their binding ability to target cells when exposed to acidic pH of 2.8 or below, a process driven by partial unfolding of S. Interestingly, HCoV-229E S is affected to a lesser extent at the same conditions, suggesting that HCoV-229E S has a higher

**Fig. 5 | SARS-CoV-2 BavPat1 RBD, NTD and S2 structure is affected by acidic pH. A–C** SARS-CoV-2 BavPat1 virions were bound to coverslips and subsequently given a pH pulse for 1 min or left untreated (pH 7.0, pH 2.6, pH 2.0, and untreated). After fixation and permeabilization, virions were stained for N as a control (cyan) and different S antibodies. In (**A**) a neutralizing RBD antibody (Cov2rbdc1-mab1, yellow) and an S2 antibody (NBP3-05701, magenta) were used, in (**B**) virions were stained with an RBD antibody (#63847, magenta) exhibiting a preference for linear epitopes, and in (**C**) virions were stained with the NTD antibody (MA5-36247, magenta). Samples were imaged with the SP8 (Leica). Shown are representative images from three to four independent replicates. Scale bar corresponds to 10 µm. **D–K** Quantification of three to four images per pH condition from three to four independent experiments each (n = 9 per group, except **D**, **E** where n = 12 for undiluted, pH 7.0 and pH 2.6 and n = 13 for pH 2.0). In brief, a background cut-off was defined for each channel and subtracted from the images. Only pixels positive for N were considered in the following analyses to further reduce background and processing artefacts. For each image, the intensity ratio of the S antibody signal to the N signal was calculated for the neutralizing RBD Cov2rbdc1-mab1 (**D**), the S2 NBP3-05701 (**F**), the RBD #63847 (linear epitope preference) (**H**) and the NTD MA5-36247 (**J**) antibody. Additionally, the Manders correlation coefficient M1 was calculated using Coloc 2 (ImageJ), measuring the fraction of the N intensity

signal associated with the neutralizing RBD Cov2rbdc1-mab1 (**E**), the S2 NBP3-05701 (**G**), the RBD #63847 (linear epitope preference) (**I**) or the NTD MA5-36247 (**K**) antibody signal. Error bars represent SD from the mean. All statistical tests were done with a nonparametric one-way ANOVA (Kruskal–Wallis test, two-sided) in GraphPad Prism with all conditions compared to the pH 7.0 control. Data was considered non-significant (ns) if p values were 0.05 or above and significant for p values below 0.05 ($p < 0.05$ (*), $p < 0.005$ (**), $p < 0.0005$ (***)). **L** WB of three different SARS-CoV-2 stock preparations (1, 2, and 3) showing N and S. Full-length S0 as well as the S2 subunit were detected with the S2 antibody. **M** Full-lengths S0 and S2 in (**L**) were quantified using Image Studio Lite and the S2 to full-length S0 ratio was calculated (n = 3). Error bars represent SD from the mean. **N** Epitopes of used antibodies visualized on S trimers with one monomer in the open conformation (PDB ID: 6ZGG). For each antibody, the structural epitope preference (linear or native) is indicated as well as the IF signal increase (↑) or decrease (↓) after treatment at pH 2.0. S domains were colored on one monomer of the trimer in the open conformation: NTD (turquois), RBD (yellow), CTD1 (dark green), CTD2 (light green), fusion peptide (orange), central helix (dark blue), heptad repeat 1 (HR1, light blue) and HR2 (purple). For each antibody, the epitope was colored in red, and overlapping domains were colored in the assigned colors. If the epitope was unknown within the S domain, the whole domain was colored red.

---

stability than SARS-CoV-2 S, resulting in retained infectivity even at extreme pH conditions.

## Discussion

Determining the physicochemical properties of IRPs and gaining knowledge on how virions are affected by these conditions are key to the development of new prevention strategies for airborne transmissible diseases. Beyond sole virus inactivation kinetics, insights into the mechanisms of inactivation are crucial to allow predictions about virus stability in IRPs and airborne transmissibility across virus strains and species. In support of global efforts to improve pandemic preparedness, this is especially important for viruses with recognized zoonotic potential like influenza A viruses and coronaviruses. In this study, we focused on the inactivation mechanism of SARS-CoV-2 at acidic pH encountered in small IRPs when exposed to acidified air compositions, achieved by raising $HNO_3$ concentrations in indoor environments. Since respective $HNO_3$ levels are considered non-hazardous, we do not expect adverse effects to human health. However, this needs to be studied in more detail before any application. Further implementation of air acidification interventions would require real-time monitoring of $HNO_3$ levels to avoid over exposure, while ensuring sufficient levels for virus inactivation. We studied two coronaviruses (SARS-CoV-2 and HCoV-229E) in a non-aerosol system to isolate the effect of acidic pH from other potential stressors in IRPs. Recently, Amruta et al. linked the inactivation of SARS-CoV-2 at acidic pH to a general disruption of virions[41]. Here we show that the viral envelope remains intact, while vRNPs are structurally altered. This aligns with the observation that HCoV-229E is stable at acidic pH of 2.0 (Fig. 1D), indicating that major structural components shared by both coronaviruses, such as the membrane, retain their functionality in promoting viral entry.

Given that IAVs are efficiently inactivated at pH values below 5.5–6.0, SARS-CoV-2 at pH of 2.8 and below and HCoV-229E is stable down to a pH of 2.0 with a 99% inactivation time of several hours[23], acidic pH affects virions in a species- and potentially strain-specific manner. The inactivation of IAV at acidic pH is due to an irreversible conformational change in its surface glycoprotein HA, more specifically the switch from the pre-fusion to the post-fusion conformation[32]. It is well established that once switched to the post-fusion conformation IAVs lose receptor binding capability[42–44]. Similarly, we found that SARS-CoV-2 attachment to target cells is disrupted after treatment with acidic pH of 2.8 or below and attributed this to conformational changes in S. Although pH resistance of these viruses is fundamentally different, the inactivation mechanism is similar regarding attachment disruption and the surface glycoprotein being the main driver of inactivation.

We were able to detect conformational changes in the SARS-CoV-2 S2 subunit induced by acidic pH treatment with LiP LC-MS of recombinant S. However, we only achieved S coverage of around 35%, therefore, we continued to study S conformation on full virions with the help of S-specific antibodies. All SARS-CoV-2 S antibodies tested showed clear differences in signal intensity and Manders correlation after treatment at pH 2.0 compared to pH 7.0 and subsequent neutralization, demonstrating structural changes in S that are irreversible (Figs. 5, 6, S4 and S6). The observation that neutralizing RBD antibodies lose binding to S, while antibodies against linear peptides gain binding, suggests unfolding of S domains, including the RBD (Figs. 5, 6 and S4). This explains the lack of bound virions on the surface of target cells (Fig. 3) and fits well to the modeling data from Niu et al. predicting major conformational changes in the RBD and S2 domain[45]. Due to virus titer restrictions, the number of virions was too low to observe quantifiable changes of S structure or number in cryo-ET. Furthermore, all cryo-ET samples were UV inactivated. Although the inactivation was performed under mild conditions to preserve overall virion structure, we acknowledge that subtle conformational changes or local damage to S or envelope components cannot be fully excluded. While our data suggests S unfolding as the most likely inactivation mechanism, we acknowledge that shedding of the S1 subunit might be an additional mechanism contributing to decreased binding of SARS-CoV-2 virions. Although NTD signals as well as signal for the RBD antibody with a linear epitope preference were increased in samples treated at pH 2.0 indicating that a high number of S1 is still present on virions, we cannot exclude the disruption of S trimers by shedding one or two out of the three S1 subunits. Trimeric S may consist of a mix of cleaved S1/2 and uncleaved S0 monomers, allowing such partial S1 shedding. Of note, SARS-CoV-2 stocks with higher amounts of cleaved S2 (Fig. 5M) did not differ in their inactivation kinetics from the stock with a majority of uncleaved S0 (Figs. 1B and S1E).

We show that HCoV-229E S has a higher resistance to unfolding events induced by acidic pH explaining the stability of the virus (Fig. 6I). In contrast, we found similar inactivation patterns of BavPat1, an early isolate, and JN.1, an Omicron isolate, at acidic pH below 3 (Figs. S1 and 6), in accordance with studies measuring similar aerosol stability of early isolates and Omicron variants[46,47]. The results from this in vitro study emphasize that investigating IRP stability of SARS-CoV-2 variants in varying air compositions influencing pH is an important aspect of future research. Future experiments should include studying pH effects within the more complex context of respiratory aerosol particles and determine if other factors might dampen or enhance acidic inactivation of coronaviruses, such as mucus components or environmental factors. Beyond acidic pH, the effect of alkalinity, salt concentrations, crystallization, ionic strength and

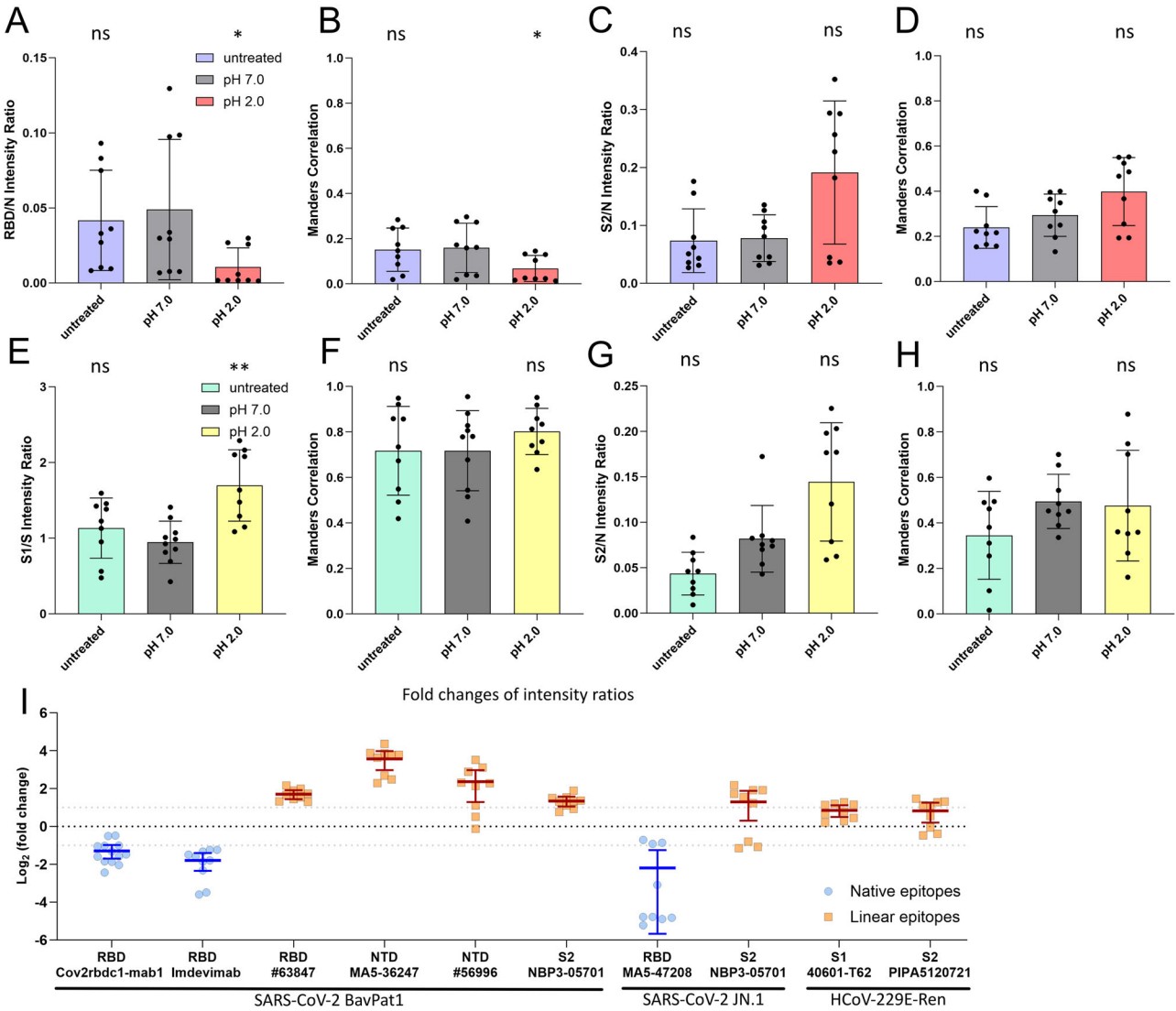

**Fig. 6 | Structural changes of SARS-CoV-2 JN.1 S and HCoV-229E S.** SARS-CoV-2 JN1 virions (**A**–**D**) or HCoV-229E-Ren virions (**E**–**H**) were bound to coverslips and subsequently given a pH pulse for 1 min or left untreated (pH 7.0, pH 2.0, and untreated). As in Fig. 5D–H, the intensity ratio of the neutralizing SARS-CoV-2 JN.1 RBD MA5-47208 (**A**), the SARS-CoV-2 S2 NBP3-05701 (**C**), the HCoV-229E S1 40601-T62 (**E**) and the HCoV-229E S2 PIPA5120721 (**G**) antibody signal to the corresponding N signal was calculated for each image. Additionally, the Manders correlation coefficient M1 was calculated for the neutralizing SARS-CoV-2 JN.1 RBD MA5-47208 (**B**), the SARS-CoV-2 S2 NBP3-05701 (**D**), the HCoV-229E S1 40601-T62 (**F**) and the HCoV-229E S2 PIPA5120721 (**H**) antibody signal. Error bars represent SD from the mean of three images derived from three independent experiments (n = 9 per group). All statistical tests were done with a nonparametric one-way ANOVA (Kruskal-Wallis test, two-sided) in GraphPad Prism with all conditions compared to the pH 7.0 control. Data was considered non-significant (ns) if p values were 0.05 or above and significant for p values below 0.05 ($p < 0.05$ (*), $p < 0.005$ (**), $p < 0.0005$ (***)). **I** $\text{Log}_2$ fold changes calculated for the intensity ratios of all antibodies used in the study comparing the pH 2.0 to the pH 7.0 condition. Antibodies with a preference for native epitopes are depicted by blue circles, while antibodies with a preference for linear epitopes are depicted by orange squares. Dotted lines indicate a $\log_2$ fold change of −1 or 1. Error bars show the 95% CI of the mean for each antibody (n = 9 per group, except RBD antibodies Imdevimab where n = 10 and Cov2rbdc1-mab1 where n = 13).

other stressors in IRPs on different viruses remains an important topic of future research to understand transmissibility. Especially salt concentration, controlled by RH, was shown to play a role in virus stability in IRPs[48,49]. The inactivation mechanism of respiratory viruses at high salt concentration as encountered in IRPs is currently unknown. A recent study by Schaub et al. suggests a more species-independent inactivation mechanism of disrupting virion structure due to supersaturated salt concentration[50]. However, virus strain-specific salt stabilities have been observed for IAV[51], raising questions about the exact mechanism of inactivation and viral strategies to compensate for the damaging effects of elevated salt concentration.

In summary, this study reveals that inactivation of SARS-CoV-2 by acidic conditions occurs through partial denaturation of S structure and a resulting defect in viral attachment. These findings contribute to a better understanding of the physicochemical processes in IRPs, which is a

prerequisite for developing intervention strategies for airborne transmission of respiratory viruses.

## Methods
### Biophysical modeling
The respiratory aerosol model ResAM used in this work is a biophysical model to determine virus inactivation in IRPs after exhalation as a function of air composition. In brief, ResAM is based on a spherical shell diffusion model, which embodies the thermodynamic and kinetic properties of respiratory fluids composed of $H_2O$, $H^+$, $OH^-$, $Na^+$, $Cl^-$, $CO_2(aq)$, $HCO_3^-$, $NH_3(aq)$, $NH_4^+$, $CH_3COOH(aq)$, $CH_3COO^-$, $CH_3COONH_4(aq)$, $NO_3^-$, as well as two classes of organic compounds with low and high molecular weight, representative of the lipids and proteins in the lung fluid. For further details on ResAM we refer to Luo et al.[23].

## Cell lines and virus stocks

Huh-7 cells (kindly provided by Georg Kochs, Universitätsklinikum Freiburg, Germany) and Vero-E6 cells (kindly provided by Volker Thiel, University of Bern, Switzerland) were cultured in complete DMEM consisting of DMEM (41966-029, Gibco) supplemented with 10% Fetal Bovine Serum (FBS, 10270-106, Gibco) and 1% Penicillin-Streptomycin 10,000 U/ml (P/S, 15140-122, Gibco). Vero-E6 cells stably expressing ACE2 and TMPRSS2 (VAT), kindly provided by Sam Wilson (University of Glasgow, United Kingdom)[52], were cultured in complete DMEM with 2 mg/ml Geneticin (10131-035, Gibco) and 200 µg/ml Hygromycin (10687010, Thermo Fisher Scientific). Experiments were conducted with the SARS-CoV-2 strain BetaCoV/Germany/BavPat1/2020 (BavPat1)[36], provided by the European Virus Archive GLOBAL (EVAg; Ref-SKU: 026V-03883), the SARS-CoV-2 JN.1 Omicron strain SARS-CoV-2/human/Switzerland/ZH-UZH-IMV15/2024 (JN.1) isolated at the Institute of Medical Virology, University of Zurich (IMV, UZH) and HCoV-229E-Ren as well as HCoV-229E WT, kindly provided by Volker Thiel (University of Bern, Switzerland)[37]. All experiments involving infectious SARS-CoV-2 were carried out in a biosafety level 3 (BSL3) facility at the IMV, UZH. Prior to the project, procedures were approved and assessed by the Swiss Federal Office of Public Health (Ecogen number A202808/3). BavPat1 and JN.1 were propagated in VAT cells and Vero-E6 cells, respectively at an MOI of 0.001 and in presence of 0.5 µg/ml TPCK Trypsin (T1426-50M, Sigma-Aldrich). Supernatants were collected when cytopathic effects (CPE) were observed for the majority of cells and cleared from cell debris by centrifugation. Subsequently, the supernatant was frozen in aliquots at −80 °C or concentrated with Amicon Ultra-15, PLHK Ultracel-PL Membrane, 100 kDa tubes (UFC910024, Milipore) by following the manufacturer's instructions. Sequencing of SARS-CoV-2 BavPat1 and JN.1 supernatants revealed that our BavPat1 stock harbors the D614G mutation and an intact furin cleavage site, while our JN.1 isolate lost the furin cleavage site in the isolation process. HCoV-229E-Ren and HCoV-229E WT were propagated on Huh-7 cells at 33 °C, without trypsin and an MOI of 0.001 or 0,01, respectively. Supernatant was harvested after 72 h and either frozen at −80 °C or concentrated by ultracentrifugation or with Amicon Ultra-15, PLHK Ultracel-PL Membrane, 100 kDa tubes. Ultracentrifugation was performed through a 30% sucrose (84100, Fluka Analytika) cushion at $112,400 \times g$ for 90 min at 4 °C with the AH-629 rotor of a Sorvall WX 100 Ultra centrifuge (Thermo Fisher Scientific). Pellets were reconstituted in 300 µl of PBS.

## Virus isolation

SARS-CoV-2 JN.1 was isolated on Vero-E6 cells from an anonymized patient sample, diagnosed SARS-CoV-2 positive by the IMV, UZH in September 2024 via PCR. Sequencing of the patient sample revealed that the patient was infected with the JN.1 Omicron variant (GenBank PV414078.1). First, the patient sample was sterile filtrated with a 0.45 µm filter and subsequently used for a 2-fold dilution series in DMEM. Vero-E6 cells were then infected by mixing $2.5 \times 10^5$ cells (suspended in 500 µl complete DMEM additionally supplemented with 2.5 µg/ml Amphotericin B (15290018, Thermo Fisher Scientific)) with the sample dilutions in a 24-well plate. Infected cells were incubated at 37 °C and monitored for CPE daily. If CPE was detected the supernatant was harvested, cleared from cell debris by centrifugation and subsequently used for standard plaque assay on Vero-E6 cells. After 4–5 days, single plaques were picked with a pipette tip and diluted in 200 µl DMEM used for infection of Vero-E6 cells. For that purpose, $5 \times 10^5$ Vero-E6 cells were seeded per well of a 6-well plate one day in advance. Infection was allowed to proceed for 1 h at 37 °C with occasional shaking to distribute the inoculum. After the completion of 1 h, 2 ml of complete DMEM was added to the well and the infected cells were incubated at 37 °C until CPE was detected. Supernatant was collected, cleared from cell debris by centrifugation, aliquoted, and frozen at −80 °C. The obtained Omicron isolate was named SARS-CoV-2/human/Switzerland/ZH-UZH-IMV15/2024.

## Virus titration: plaque assay and foci assay

SARS-CoV-2 titers were determined by standard plaque assay on VAT cells reaching a detection limit of 10 plaque-forming units (PFU)/ml. Of note, no trypsin was added to the overlay and the virus was allowed to replicate for ~44 h before fixation and plaque detection. HCoV-229E-Ren was titrated by foci assay on Huh-7 cells. Cells were seeded 1 day in advance on poly-lysine (P4832, Sigma-Aldrich) coated 12-well plates to reach 100% confluency on the day of infection. Cells were infected with HCoV-229E in 100 µl PBSi consisting of DPBS (14190-094, Gibco) supplemented with 0.2% BSA Fraction V (1ET4.3, Roth), 1% P/S, 1 mM CaCl₂ (102378, Sigma-Aldrich), and 1 mM MgCl₂ (M2670, Sigma-Aldrich) and incubated at 33 °C for 1 h. Virus was allowed to replicate for ~72 h at 33 °C before fixation with 3.7% formaldehyde (47608, Sigma-Aldrich) in PBS. Cells were blocked and permeabilized for 1 h at RT with confocal buffer consisting of 50 mM $NH_4Cl$ (31107, Sigma-Aldrich), 0.1% Saponin (47036, Sigma-Aldrich) and 2% BSA (A7906, Sigma-Aldrich) in PBS, followed by incubation with a rabbit anti-N antibody (40640-T62, Sino Biological) in confocal buffer for 1–2 h (Table 1). Cells were washed with PBS and then incubated with an anti-Rabbit Peroxidase antibody (A0545, Sigma-Aldrich) for 1 h at RT (Table 1). Again, cells were washed with PBS and subsequently $H_2O$ before addition of TrueBlue Peroxidase Substrate KPL (5510-0030, Seracare) for 15 min at RT. After removing the substrate, foci were counted by eye to determine the titer in foci-forming units (FFU)/ml.

## Acidification of viruses

Milli-Q $H_2O$ was adjusted to pH 7.1 or pH 2.6 with the disodium phosphate (71645 Sigma-Aldrich) and citric acid (C1909 Sigma-Aldrich) buffer system described by Sigma-Aldrich[53] and measured with the Ultra-Micro-ISM electrode and FiveEasy pH meter (Mettler Toledo). pH 2.0 was reached in a 0.1 M citric acid solution. SARS-CoV-2 was spiked into neutral (pH 7.1) or acidified water in a 1:20 dilution and the pH was measured with the Orion™ 9810BN Micro pH Electrode (9810BN, Thermo Fisher Scientific) and Orion™ Versa Star Pro™ pH/LogR™ Benchtop Meter (VSTAR81, Thermo Fisher Scientific). The reported pH has an inaccuracy of ±0.1 units due to variation between different replicates. Neutralization was performed by diluting samples 1:100 in PBSi additionally buffered with 3.5 mM citric acid and 33 mM disodium phosphate (pH 7 PBSi). Sample dilution allows neutralization without using strong bases, potentially affecting virus titer. To obtain samples with higher virus titers for attachment assays and cryo-ET, the reaction titer was increased by spiking SARS-CoV-2 into acidified water in a 1:2 dilution. In this case, a pH of 2.8 was reached in a reaction containing 0.2 M citric acid. For HCoV-229E, concentrated stock was directly acidified to achieve the highest possible titer. 10× pH 7 buffer (176 mM citric acid and 165 mM disodium phosphate) or 1 M citric acid was diluted 1:10 in HCoV-229E stock reaching a reaction pH of 7.1 and 2.0, respectively. To test the effects of different concentration methods and virus titers during the pH reaction, HCoV-229E-Ren stocks were spiked into neutral or acidified water in a 1:10, 1:20, or 1:100 dilution as described for SARS-CoV-2. To analyze S conformation on SARS-CoV-2 BavPat1, JN.1 and HCoV-229E virions via IF, $10^7$ PFU, $5 \times 10^5$ PFU, or $7.5 \times 10^5$ FFU, respectively, were centrifuged onto cover slips in 200 µl PBSi for 30 min at $1000 \times g$ and 4 °C. Subsequently, cover slips were washed and 300 µl buffered water at pH 7.1, pH 2.6, or pH 2.0 was added. After 1 min incubation, the buffered water was removed, and cover slips were washed three times to neutralize the virus. Virions on cover slips were fixed with 4% paraformaldehyde (PFA, 15710, Lucerna) in PBS prior to IF staining.

## Plunge freezing

SARS-CoV-2 was UV inactivated with the UV Analysis Lamp UV-4 S (29 50 400, Herolab) for 10 min in 24-well plates. A maximum of 600 µl were inactivated at a time and inactivation efficiency was validated by plaque assay. Additionally, we tested the inactivation in outgrowth assays by incubating inactivated SARS-CoV-2 samples on VAT cells for 7 days. No CPE was observed in outgrowth assays for inactivated samples and plaque assays of the retrieved supernatants were negative. Holey carbon

**Table 1 | Antibody summary including dilutions for different assays**

| Antibody | WB | IF | Foci |
|---|---|---|---|
| Human coronavirus (HCoV-229E) Nucleocapsid Antibody, Rabbit PAb, Antigen Affinity Purified, 40640-T62 Sino Biological, RRID:AB_3676912 | | | 1:10,000 |
| Anti-Rabbit IgG (whole molecule)–Peroxidase antibody produced in goat, A0545, Sigma-Aldrich, RRID:AB_257896 | | | 1:10,000 |
| SARS/SARS-CoV-2 Nucleocapsid Monoclonal Antibody, MA5-29981, Thermo Fisher Scientific, RRID:AB_2785780 | 1:1000 | 1:1000 | |
| SARS-CoV-2 Spike RBD monoclonal human IgG1 antibody (Clone H4), cov2rbdc1-mab1, Invivogen | 1:1000 | 1:100 | |
| Imdevimab (Synonyms: REGN10987); HY-P99342, Medchem Express, RRID:AB_3695156 | 1:1000 | 1:2000 | |
| SARS-CoV-2 Spike S2 Antibody - BSA Free, NBP3-05701, Novus Bio, RRID:AB_3534207 | 1:1000 | 1:400 | |
| SARS-CoV-2 Spike Protein (RBD) (E7B3E) Rabbit mAb, #63847 Cell Signaling Technology, RRID:AB_3674090 | 1:1000 | 1:200 | |
| SARS-CoV-2 Spike Protein S1 Recombinant Rabbit Monoclonal Antibody (HL6), MA5-36247, Thermo Fisher Scientific, RRID:AB_2890589 | 1:1000 | 1:50 | |
| SARS-CoV-2 Spike Protein (S1-NTD) Antibody, #56996, Cell Signaling, RRID:AB_3492098 | 1:1000 | 1:200 | |
| SARS-CoV-2 Spike Protein RBD Omicron Recombinant Rabbit Monoclonal Antibody (HL1867), MA5-47208 Thermo Fisher Scientific, RRID:AB_2938280 | 1:1000 | 1:500 | |
| Human coronavirus (HCoV-229E) Nucleocapsid Antibody, Mouse Mab, 40640-MM11, Sino Biological, RRID:AB_3676911 | 1:1000 | 1:400 | |
| Human coronavirus (HCoV-229E) Spike S1 Antibody, Rabbit PAb, Antigen Affinity Purified, 40601-T62 Sino Biological | 1:1000 | 1:100 | |
| HCoV-229E Spike S2 Polyclonal Antibody, PIPA5120721, Invitrogen, RRID:AB_2914293 | 1:1000 | 1:100 | |
| GAPDH Antibody (FL-335), sc-25778, Santa Cruz, RRID:AB_10167668 | 1:2000 | | |
| IRDye anti-rabbit 800CW, 926-32211, Li-Cor, RRID:AB_621843 | 1:10,000 | | |
| IRDye anti-mouse 680RD, 926-68070, Li-Cor, RRID:AB_10956588 | 1:10,000 | | |
| Donkey anti-Mouse IgG (H + L) Highly Cross-Adsorbed Secondary Antibody, Alexa Fluor™ 488, A-21202; Thermo Fisher Scientific, RRID:AB_141607 | | 1:1000 | |
| Goat anti-Human IgG (H + L) Cross-Adsorbed Secondary Antibody, Alexa Fluor 594, A-11014, Thermo Fisher Scientific), RRID:AB_2534081 | | 1:1000 | |
| Donkey anti-Rabbit IgG (H + L) Highly Cross-Adsorbed Secondary Antibody, Alexa Fluor™ 647, A-31573, Thermo Fisher scientific, RRID:AB_2536183 | | 1:1000 | |

grids (Cu 200 mesh, R2/1, Quantifoil®) were plasma-cleaned for 10 s in a Gatan Solarus 950 (Gatan). Samples were mixed (1:10) with 10× concentrated 10 nm protein A gold (Aurion) prior plunge freezing. Stock samples (3 µl) were applied on a grid for plunge freezing into liquid ethane using an automatic plunge freezer EM GP2 (Leica). The ethane temperature was set to −183 °C and the chamber to room temperature with 80% humidity. Grids were blotted from the back with Whatman™ Type 1 paper for 2.5 s. Neutralized samples (3 µl) were applied on a grid for plunge freezing into liquid ethane using a Vitrobot Mark IV (Thermo Fisher Scientific) with 5 s blotting time and room temperature with 99% humidity in the chamber. Grids were clipped into AutoGrids™ (Thermo Fisher Scientific).

## Cryo-electron tomography (cryo-ET) and tomogram reconstruction

Cryo-ET was performed using a Krios cryo-TEM (Thermo Fisher Scientific) operated at 300 keV and equipped with a post-column BioQuantum Gatan Imaging energy filter (Gatan) and K3 direct electron detector (Gatan) with an energy slit set to 15 eV. As a first step, positions on the grid were mapped at 8700× (pixel spacing of 10.64 Å) using a defocus of ∼−65 µm in SerialEM[54] to localize SARS-CoV-2 virions. Tilt series were acquired using a dose-symmetric tilting scheme[55] with a nominal tilt range of 60° to −60° with 3° increments with SerialEM. Due to the higher concentration of the stock sample, tilt series were acquired using PACEtomo[56] (version 1.6.1) in SerialEM (version 4.1.0 beta). Tilt series were acquired at target defocus range of −3 to −4 µm, with an electron dose per record of 3 e⁻/Å² and a magnification of ×33,000 (pixel spacing of 2.671 Å). Beam-induced sample motion and drift were corrected using MotionCor2[57] (version 1.3.1). Tilt series were aligned using the fiducial model in IMOD[58] (version 5.0.1). Tomograms were reconstructed using R-weighted back projection algorithm with dose-weighting filter and SIRT-like filter 5 in IMOD[58]. In IMOD, 12 slices of the final tomogram were averaged for visualization.

## Measurements in tomograms

Diameter and spikes per virion were measured manually in IMOD[58]. One contour with two points was used to measure the diameter on one axis. Two axes were measured (long/a and short/b), and the mean was used as the mean diameter per virion. Spikes per virion were counted manually in IMOD using the object category in the model to separate pre- and postfusion spikes. Points were used to count individual spikes. The sphericity (eccentricity) was calculated by using the formula shown below.

$$e = \frac{\sqrt{(a^2 - b^2)}}{a}$$

## RNase digestion assay and qPCR

For the RNase digestion assay 5 µl 10× RNA digestion buffer consisting of 1 mM Tris-EDTA (T9285, Sigma-Aldrich) and 0.5 M NaCl (71380, Sigma-Aldrich) (pH 7 to 7.5) were added to 45 µl of neutralized virus samples or controls (RNA extracted from a SARS-CoV-2 stock, a SARS-CoV-2 sample and a SARS-CoV-2 stock sample inactivated with 80% ethanol for 10 min). Subsequently, 1 µl RNase A/T1 mix (EN0551, Thermo Fisher Scientific) or 1 µl water (undigested control) was added and incubated for 30 min at 37 °C. Digestion was terminated by adding 0.5 µl SUPERase-In RNase Inhibitor (AM2694, Thermo Fisher Scientific) to each sample and incubation for 20 min at RT. RNA extraction was performed using the ReliaPrep RNA Cell Miniprep System (Z6012, Promega) following the manufacturer's protocol. Extracted RNA was reverse transcribed by SuperScript IV (18090050, Thermo Fisher Scientific) using Random Primers (C1181, Promega). Primers (Microsynth) for the SARS-CoV-2 E gene (fw: ACAGGTACGTTAATAGTTAATAGCGT; rv: ATATTGCAGCAGTACGCACACA) were used in combination with PowerTrack SYBR Green Master Mix (A46112, Thermo Fisher Scientific) for qPCR performed with the 7300 Real time PCR

System (Applied biosystems). Genomic copy numbers were determined with the 7300 System SDS Software (version 1.4) by preparing a standard curve from the pGBW-m4252984 vector (153898, Addgene). Results were analyzed as follows: First, genomic copies (GC) after RNase digestion were divided by GC measured without RNase digestion of the same sample, resulting in the fraction of maintained genomes (c). Subsequently, c was normalized to c0 (c of the 10 s pH 7 sample) from the same experiment. Data was logarithmically transformed showing $\log_{10}$ of GC loss of each sample relative to the 10 s pH 7 sample. Additionally, this calculation was performed on the infectious titers as determined by plaque assay from the same samples. The correlation between $\log_{10}$ PFU reduction and $\log_{10}$ GC reduction was calculated with GraphPad Prism (version 10.2.3) using a linear regression model.

### Infection and attachment assay
VAT cells were seeded onto cover slips one day prior to infection. If samples were used for super-resolution microscopy, cover slips were cleaned with 100% ethanol for 5 min beforehand. The next day, VAT cells were either infected with 100 μl of neutralized samples for 1 h at 37 °C or pre-cooled for 30 min at 4 °C and then infected with 200 μl of neutralized samples for 1.5 h on frozen metal plates to allow binding, but inhibit internalization[39]. For single cycle infections, cells were incubated at 37 °C for 6 h and subsequently fixed with 3.2% PFA and stained for N by IF (see below). For binding assays, cells with bound virions were washed three times with cold PBS and fixed directly after with 3.2% PFA prior to IF staining.

### Limited proteolysis (LiP) liquid chromatography-mass spectrometry (LC-MS)
Recombinant trimeric S (40589-V08H8, Sino Biological) was diluted in H2O according to the manufacturer's recommendations and split into two equal portions of 50 μg. The trimers were then treated for 2 min at pH 2.0 or 7.0 by adding 1 M citric acid or 10× pH 7 buffer in a 1:10 ratio to the solution as described in section "acidification of viruses". Subsequently, the solutions were neutralized by diluting samples 1:20 in PBS additionally buffered with 3.5 mM citric acid and 33 mM disodium phosphate. We used Amicon tubes (UFC5050) to concentrate S trimers and rebuffer to LiP buffer (20 mM HEPES pH 7.4, 150 mM KCl, and 10 mM MgCl2). Resulting S concentrations were measured by NanoDrop (NanoDrop One^C, Thermo Fisher Scientific) and BCA assay (Pierce™ BCA Protein Assay Kits, 23227 Thermo Fisher Scientific). Samples were adjusted to the same protein concentration. We performed LiP-MS in four analytical replicates per sample, using 3 μg of S per replicate supplemented with 7.5 μg of BSA. LiP was essentially executed as described by Malinovska et al.[59] with minor modifications. In brief, all samples were digested by proteinase K (MC505, Promega) for 5 min at 25 °C at an enzyme to protein ratio of 1:100 (w/w). Next, proteins were denatured at 99 °C and adjusted to 5% sodium deoxycholate (SDC), followed by reduction, alkylation, and full tryptic digestion (V5113, Promega) at 1% SDC. Afterwards, SDC was removed by acid precipitation and MW filtering. Remaining peptides were desalted on C18 StageTips. LC-MS analysis of purified peptides was performed with the ACQUITY UPLC M-Class System (Waters) directly connected to an Orbitrap Exploris™ 480 Mass Spectrometer (Thermo Fisher Scientific) operated in data independent acquisition mode. The resulting chromatographic data was analyzed by using directDIA workflow in Spectronaut (19.9.250422.62635) allowing for semi-tryptic peptides. Results for all identified peptides are provided in Supplementary Data 1 and raw data are available via ProteomeXchange with identifier PXD064612.

### Immunofluorescent (IF) staining
After fixation and washing, samples were blocked and permeabilized with confocal buffer for 1 h at RT and subsequently stained with the primary antibodies for 2 h in confocal buffer. Primary antibodies against SARS-CoV-2 antigens are anti-N (MA5-29981, Thermo Fisher Scientific), anti-RBD (cov2rbdc1-mab1, Invivogen), Imdevimab (HY-P99342, Medchem Express), anti-RBD (#63847, Cell Signaling Technologies), anti-NTD

(MA5-36247, Thermo Fisher Scientific), anti-NTD (#56996, Cell Signaling), and anti-S2 (NBP3-05701, Novus Bio) (Table 1). Only the anti-N and anti-S2 SARS-CoV-2 antibodies were able to bind to SARS-CoV-2 JN.1, therefore, we selected an additional RBD antibody (MA5-47208, Thermo Fisher Scientific) neutralizing Omicron variants. For staining of HCoV-229E virions, we used anti-N (40640-MM11, Sino Biological), anti-S1 (40601-T62, Sino Biological), and anti-S2 (PA5120721, Invitrogen) antibodies (Table 1). Samples were washed and stained with secondary antibodies and DAPI (10236276001, Sigma-Aldrich) for 1 h. The following secondary antibodies were used: anti-Mouse Alexa 488, anti-Human Alexa 594, and anti-Rabbit Alexa 647 (A-21202, A-11014, and A-31573, Thermo Fisher Scientific) (Table 1). Of note, if anti-Human Alexa 594 was used, staining was performed prior to a second round of primary and secondary antibody staining, since unspecific binding of anti-Human Alexa 594 to our SARS-CoV-2 N antibody was observed. For virus attachment assays, cells were washed with PBS after completion of the IF staining and additionally stained with a 1:2000 dilution of CellBrite® NIR680 (#30070, Biotium) in PBS. Finally, cover slips were mounted with ProLong® Gold Antifade Mountant (P36930, Thermo Fisher Scientific).

### Fluorescent imaging, image processing and analysis
For images of single-cycle infections, the DMi8 microscope (Leica) was used in combination with the inbuilt THUNDER Instant Computational Clearing Algorithm (ICC, Leica). Images were exported from the LAS X software (version 3.0.2.16120, Leica). To image virions on cover slips in the absence of cells, the SP8 confocal laser scanning microscope (Leica) was used. Images were imported into ImageJ (version 2.14.0; Java 1.8.0_172) and background was subtracted. The inbuilt Coloc 2 package was used to calculate intensity sums and Manders correlation for each image using the N channel as a mask (area of interest). Three to four images per experiment were quantified and plotted in GraphPad Prism. To calculate fold changes in intensity ratios all values of the pH 2.0 condition were normalized to the average of the pH 7.0 condition. The mean as well as upper and lower limits (determined by the 95% confidence interval) of each data set were calculated with GraphPad Prism (version 10.2.3) and $\log_2$ transformed. For virus attachment assays, a super-resolution approach was chosen to obtain a better z-resolution. Samples were imaged with the ELYRA 7 (Zeiss) in combination with the ZEN software (black edition 3.0, Zeiss). Lattice SIM² Processing was used for deconvolution (Zeiss). A high signal-to-noise ratio (SNR) approach was used for the membrane staining with 25 iterations and a regularization weight of 0.007, while a medium SNR approach with 20 iterations and a regularization weight of 0.2 was applied for the N staining of virions. The resulting data was imported into Bitplane Imaris (version 10.1.1) to create maximal projections and extended x-z- and y-z-projections. Figures were assembled with Affinity Designer 2 (version 2.4.2).

### Western blot and band quantification
VAT cells were infected with an MOI of 2 of SARS-CoV-2 BavPat1 or JN.1 and lysed at 16 or 20 hpi, respectively, with Laemmli Sample Buffer (1610747, Bio-Rad) supplemented with 10% β-mercaptoethanol (BME, M6250, Sigma-Aldrich). Huh-7 cells were infected with HCoV-229E WT at an MOI of 4 and lysed with Laemmli Sample Buffer containing BME after 28 hpi. SARS-CoV-2 BavPat1 stock samples were lysed 1:2 in Laemmli Sample Buffer containing BME. Samples were heated to 95 °C for 5–10 min before loading on a Bolt 4–12% Bis-Tris 1.0 mm gel (NW04120BOX, Thermo Fisher Scientific). The Precision Plus Protein™ All Blue Prestained Protein Standards (#1610373, Bio-Rad) were used to determine protein sizes. Proteins were separated by PAGE and transferred to a 0.45 μm nitrocellulose membrane (10600008, Amersham) according to the manufacturer's instructions. Afterwards, the membrane was blocked in Tris-buffered saline (TBS) containing 0.1% Tween20 (93773, Sigma-Aldrich) (TBS-T) and 5% milk (T145.3, Roth). Proteins were incubated with primary antibodies binding S, N, or GAPDH in TBS-T with 5% milk at 4 °C overnight (Table 1). Membranes were washed and secondary antibodies, IRDye anti-rabbit 800CW and IRDye

anti-mouse 680RD (926-32211 and 926-68070, Li-Cor), were applied to the membrane in TBS-T with 5% milk and incubated for 1 h at RT (Table 1). Images were generated with the Odyssey Fc Imager (Li-Cor) and the signal was quantified with Image Studio Lite (version 5.2, Li-Cor). All uncropped WBs are provided in Fig. S8.

## Statistics and reproducibility

All statistical tests concerning microscopy images (IF and cryo-ET quantification) as well as linear regression for RNase digestion assay data were performed in GraphPad Prism (version 10.2.3). For cryo-ET data, the nonparametric Mann-Whitney test (two-tailed) was used, and for IF-based data, we conducted nonparametric one-way ANOVA (Kruskal–Wallis test, two-sided) comparing all conditions to the pH 7.0 control. Data was considered non-significant (ns) if p values were 0.05 or above and significant for p values below 0.05 ($p < 0.05$ (*), $p < 0.005$ (**), $p < 0.0005$ (***)). Exact p values and 95% confidence intervals are provided in Supplementary Data 2. For LiP LC-MS statistics were performed in Spectronaut (19.9.250422.62635), and peptides with a higher negative $\log_{10}$ p value than 2.439 (corresponding to a Q value of 0.05) were considered significant.

## Reporting summary

Further information on research design is available in the Nature Portfolio Reporting Summary linked to this article.

## Data availability

All data, except for the raw LiP-MS data, supporting the findings of this study are available within the paper and its supplementary information files. Uncropped blots are provided in Fig. S8. LiP LC-MS data have been deposited to the ProteomeXchange Consortium via the PRIDE[60] partner repository with the dataset identifier PXD064612. Results for all peptides identified in LiP LC-MS are provided in Supplementary Data 1. The sequence of the SARS-CoV-2 JN.1 isolate is available on GenBank (PV414078.1). Source data of graphs presented in the figures as well as exact p values are available in Supplementary Data 2. Any additional data is available from the corresponding author upon reasonable request.

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

## Acknowledgements

This study was funded by the Swiss National Science Foundation (grant number 189939 to S.St., T.K., U.K.K., A.N.) and the UZH Candoc Grant (grant number FK-23-037 to I.G.). This work was further supported by a research grant from the Chica and Heinz Schaller Foundation (Schaller Research Group Leader Programme) to P.C. and by Flagship Initiative-Engineering Molecular Systems, Project ExU 6.1.20_CoVLP to P.C. and L.Z. A.G.W. is supported by the Wellcome Trust award 303026/Z/23/Z. We thank the Center for Microscopy and Image Analysis (ZMB, UZH) and especially Nicolas Schilling, Johannes Riemann, and Jana Döhner for the assistance with fluorescent imaging. We thank the diagnostics at the IMV, UZH, and especially Gabriela Ziltener, Claudio Pfister, Selina Leiber, and Daniel Frei for assisting with SARS-CoV-2 variant isolation and sequencing. We thank the Infectious Diseases Imaging Platform (IDIP) at the Center for Integrative Infectious Disease Research Heidelberg and the cryo-EM network at Heidelberg University (HD-cryoNET) for support and assistance. The authors gratefully acknowledge the data storage service SDS@hd supported by the Ministry of Science, Research, and the Arts Baden-Württemberg (MWK), the German Research Foundation (DFG) through grant INST 35/1314-1 FUGG and INST 35/1503-1 FUGG. We also thank the Functional Genomics Center Zurich (FGCZ, UZH) and especially Tobias Kockmann for his assistance with LiP LC-MS.

## Author contributions

Conceptualization by I.G., S.St., T.K., A.G.W., T.P., U.K.K. Methodology by I.G., L.Z., P.C., B.L., A.G.W. Software by B.L. Formal analysis by I.G., L.Z., M.H. Investigation by I.G., L.Z., B.L., M.O.P., A.G.W., E.G. Resources by S.St., P.C., T.K., U.K.K., T.P., A.N. Writing—original draft preparation by I.G., L.Z., S.St. Writing—review editing by I.G., S.St., L.Z., P.C., T.K., N.B., M.O.P., L.K.K., A.G.W., A.S., B.L., S.C.D., N.B., A.N., U.K.K., M.H., T.P. Visualization by I.G., L.Z., B.L. Supervision by S.St., P.C., T.K., T.P., U.K.K. Project administration by S.St. Funding acquisition by S.St., P.C., T.K., T.P., U.K.K., A.N., I.G., A.G.W.

## Competing interests

The authors declare no competing interests.
