## [Transparent Peer Review file · Communications Biology]

Inactivation of SARS-CoV-2 at acidic pH is driven by partial unfolding of spike

Corresponding Author: Professor Silke Stertz

Version 0:

Reviewer comments:

Reviewer #1

(Remarks to the Author)

This study first clarifies that the inactivation mechanism of SARS-CoV-2 at pH below 3 is not caused by physical disruption of the viral particle structure (such as membrane rupture or genome leakage), but rather by irreversible conformational changes in the spike protein. Through super-resolution microscopy and immunofluorescence experiments, it was found that acidic pH induces partial unfolding of the S protein, exposing linear epitopes and disrupting its native conformation, thereby blocking the virus's ability to bind to host cells. This finding provides a new perspective for understanding the stability of coronaviruses in aerosol environments. However, there are several issues should be considered before publication.

1. The study found that the main cause of virus inactivation under low pH conditions is the disruption of the correct conformation of the spike protein, which prevents the virus from binding to the cell surface receptors. However, there is a lack of direct molecular evidence to prove the extent of this conformational change.
2. From Fig. 2J, it can be observed that after acidic treatment, there is a noticeable reduction in the spike abundance on SARS-CoV-2 virions. It is unclear whether this reduction in spike density contributes to a decrease in the virus's ability to invade host cells. And finally, in conjunction with the conformational changes in the S, ultimately leads to the near-complete inactivation of the virus at pH below 3.

Reviewer #2

(Remarks to the Author)

1. Brief Summary

This manuscript investigates the mechanism by which acidic pH inactivates SARS CoV 2. The authors demonstrate that exposure to pH values below 3 rapidly reduces viral infectivity—by 1,000 to 10,000 fold—primarily through inducing partial unfolding of the spike (S) protein, which in turn disrupts its ability to bind to host cells. Notably, the viral envelope and genomic RNA remain intact under these conditions. The study employs a combination of in vitro acid treatment, infectivity assays, cryo-electron tomography, RNase protection assays, and detailed immunofluorescence analyses using multiple S-specific antibodies. A comparison with HCoV 229E (using a Renilla luciferase reporter) underscores a species-specific sensitivity to acidification. In addition, aerosol modeling (via ResAM simulations) is used to explore the potential for environmental acidification as a decontamination strategy.

2. Overall Impression

The manuscript addresses an important and timely question regarding SARS CoV 2 stability and environmental inactivation. The multi-pronged experimental approach is a strength, with complementary techniques supporting the central hypothesis that low pH induces conformational changes in the spike protein leading to loss of receptor binding. However, several aspects of the study—such as the choice of viral isolates, the indirect nature of some measurements, and limitations in the structural data—warrant further clarification and discussion.

3. Specific Comments

1. Control Virus for HCoV 229E:

o The HCoV 229E strain used encodes a Renilla luciferase reporter (HCoV 229E-Ren), while BavPat1 (SARS CoV 2) does not. The manuscript should clarify whether the reporter virus serves as a suitable control for comparing acid inactivation kinetics and whether the reporter might influence viral behavior relative to a wild-type strain.

2. Choice of SARS CoV 2 Isolate:

o BavPat1 is an early (wild-type) isolate, yet current transmission is predominantly driven by variants like Omicron. The

authors should justify the choice of BavPat1 for most experiments and discuss the practical implications of using a wild-type strain versus a currently circulating variant. If feasible, inclusion of additional data using the Omicron strain throughout the study would enhance relevance.

3. Impact of UV Inactivation on Cryo-ET:

o In line 178, the authors mention that UV inactivation was performed prior to cryo-electron tomography. It is important to discuss whether UV treatment could affect virion morphology or subtle structural features, thereby influencing the cryo-ET results.

4. Cryo-ET Resolution and Virion Concentration:

o The resolution of the tomograms appears insufficient to capture subtle differences in spike conformation, and the low concentration of virions further limits quantitative analysis.

5. Evidence for Attachment-Only Conditions:

o The manuscript states that neutralized SARS CoV 2 samples (treated at pH 7.1 or 2.8) were incubated with VAT cells on ice for 1.5 h to permit attachment while preventing internalization (lines 203–204). It would strengthen the study to provide evidence or references supporting that these incubation conditions reliably block internalization yet allow for efficient attachment.

6. Antibody Profiling versus Direct Imaging:

o The assessment of spike conformational changes is largely based on antibody binding profiles. Notably, antibody testing for HCoV 229E is lacking, and reliance on indirect antibody profiling (versus direct high-resolution imaging or biophysical assays) may limit the conclusiveness of the findings. The authors should discuss the limitations of this approach and, if possible, consider additional methods to directly validate conformational changes in the spike protein.

7. Evidence of S-RBD Cleavage at Low pH:

o The review raises the question of whether there is any evidence that the receptor binding domain (RBD) is cleaved under low-pH conditions. Clarification is needed on whether the observed loss of native RBD epitopes results solely from unfolding or if cleavage may also contribute.

8. Figure Discrepancy:

o In Figure 1c (lower panel), there is an inconsistency where the description indicates a pH of 2.2, but the figure appears to label it as 2.1. The authors should verify and correct this discrepancy to ensure clarity.

9. General In Vitro Limitations:

o The experiments are conducted under controlled in vitro conditions that may not fully recapitulate the complexity of natural respiratory aerosols (e.g., presence of mucus, varying protein concentrations, and other environmental factors). A discussion of these limitations and their implications for translating the findings into real-world scenarios is warranted.

10. Virus Stock Preparation Variability:

o Variability in virus concentration and stock preparation methods is noted. The authors should clearly describe how these factors were controlled across experiments and discuss their potential impact on the observed inactivation kinetics.

11. Practical Implications for Decontamination:

o Although the study shows that acidic pH effectively inactivates SARS CoV 2 in vitro, achieving such conditions safely in real-world environments poses significant challenges. A more detailed discussion on the feasibility, safety, and limitations of applying these findings to decontamination strategies is recommended.

Reviewer #3

(Remarks to the Author)

In this manuscript, Glas et al. investigate the low stability of SARS-CoV-2 under acidic conditions and its implications for virus inactivation, particularly in the context of airborne transmission. The authors demonstrate that low pH has minimal impact on the integrity of the viral membrane but induces conformational changes in the spike (S) protein, which leads to the inactivation of SARS-CoV-2. This is supported by cryo-electron tomography and various other techniques, providing a relatively comprehensive analysis. Notably, the authors find that SARS-CoV-2 is sensitive to low pH, whereas another human coronavirus, 229E, is not. These findings are significant, but the manuscript would benefit from testing a broader range of variants, particularly currently circulating strains, and exploring practical applications for decontamination strategies. Several issues need to be addressed.

1. Line 125: According to the Methods section, the Omicron variant used in this manuscript is JN.1. Therefore, "Omicron" should be replaced with "JN.1."

2. Why are Omicron variants more transmissible, considering they are more sensitive to acidic pH, even at pH 4.3 (Fig S1E)?

3. According to the cryo-EM structure of the S protein of 229E, the S proteins of 229E adopt a closed conformation, while the S proteins of the original SARS-CoV-2 strain likely exist in a mixed open and closed conformation. It would be interesting to select a SARS-CoV-2 variant with a closed conformation and analyze its sensitivity to low pH.

4. In Fig. 2J, the morphology of the S proteins in the right panel appears to differ from those in the left panel. Additionally, low pH treatment clearly caused conformational changes in the S protein, as shown in Figures 3 and 4. This seems contradictory to the statement in line 194: "no major changes in overall morphology occurred."

5. In lines 198, 219, and 280, "virion integrity" should be revised to "virion membrane integrity."

6. The authors may consider measuring ACE2 binding by various pH-treated viruses using an ELISA.

7. The effect of pH on S1 shedding should be assessed by pelleting the virions through ultracentrifugation.

8. Limited trypsin digestion could be considered to evaluate conformational changes in the S protein induced by low pH treatment.

9. In line 377, the manuscript states that BavPat1 and IMV15 were propagated in VAT and Vero E6 cells. The sequences of the SARS-CoV-2 S proteins should be verified by sequencing to rule out the possibility of partial furin site deletions.

10. In Fig. 4I, three preparations of SARS-CoV-2 show significant differences in the S1/S ratio. Additional western blot analysis using an anti-S2 antibody might be considered.

Version 1:

Reviewer comments:

Reviewer #1

(Remarks to the Author)

In this revised manuscript, the authors have effectively addressed the initial reviewer comments and incorporated the suggested changes, substantially enhancing the overall quality and clarity of the paper. However, one point worth noting is that in the application of the LiP method to assess the conformational effects of acidic pH on the S protein, the authors utilized a recombinant S protein bearing the 6P mutation, which stabilizes the protein in its prefusion conformation. While this approach offers advantages in structural consistency, testing with the native (wild-type) S protein may provide additional insights into pH-induced allosteric changes and more accurately reflect the physiological relevance of the identified allosteric sites.

Reviewer #2

(Remarks to the Author)

The revised manuscript has addressed most of the concerns I raised.

Reviewer #3

(Remarks to the Author)

All of my concerns have been addressed.

Glas et al., Response to reviewers' comments

We thank the Editor and Reviewers for their thoughtful evaluation of our manuscript and for providing constructive feedback to improve the clarity, rigor, and relevance of our study. We now substantially revised the manuscript according to the comments. Below we provide a point-by-point response to the concerns raised. As suggested, we included a broader range of virus variants for key experiments and an additional method to track conformational changes in Spike. Of note, we added two new main figures and two supplementary figures addressing these points.

Reviewer #1 (Remarks to the Author):

This study first clarifies that the inactivation mechanism of SARS-CoV-2 at pH below 3 is not caused by physical disruption of the viral particle structure (such as membrane rupture or genome leakage), but rather by irreversible conformational changes in the spike protein. Through super-resolution microscopy and immunofluorescence experiments, it was found that acidic pH induces partial unfolding of the S protein, exposing linear epitopes and disrupting its native conformation, thereby blocking the virus's ability to bind to host cells. This finding provides a new perspective for understanding the stability of coronaviruses in aerosol environments. However, there are several issues should be considered before publication.

1. The study found that the main cause of virus inactivation under low pH conditions is the disruption of the correct conformation of the spike protein, which prevents the virus from binding to the cell surface receptors. However, there is a lack of direct molecular evidence to prove the extent of this conformational change.

We acknowledge that there was a lack of direct molecular evidence for structural changes in Spike induced by low pH. To tackle this, we utilized limited proteolysis liquid chromatography-mass spectrometry (LiP LC-MS) to identify specific sites in S affected by low pH (lines 245 - 263 and new figure 4). Our new data reveal changes in accessibility of S2 and also the NTD domain of S1 upon low pH treatment, thus providing additional evidence for partial unfolding of S at pH 2. Please note that resolving the full structure of S after pH treatment and neutralization was out of scope for this project.

Figure 4: Peptide candidates determined by LiP LC-MS after pH treatment of recombinant S. A) Volcano plot showing the Log₂ fold change of all S peptides derived from the pH 2.0 sample compared to the pH 7.0 sample. Dotted lines indicate log₂ fold changes of -1 and 1 as well as the significance threshold. Peptides with significant changes above a log₂ fold change of 1 were colored orange, while peptides with significant changes below 1 were colored blue. Data was derived from 4 technical replicates per condition. B) S trimer structure (PDB ID: 6ZGG) from different angles showing the location of significant peptides. Significant peptides above a log₂ fold change of 1 were colored orange, while peptides with significant changes below 1 were colored blue. Proteinase K cleavage sites were colored red for each peptide. Of note, the peptide “SVVNIQK” is in an unresolved region of S2 and therefore not depicted.

2. From Fig. 2J, it can be observed that after acidic treatment, there is a noticeable reduction in the spike abundance on SARS-CoV-2 virions. It is unclear whether this reduction in spike density contributes to a decrease in the virus's ability to invade host cells. And finally, in conjunction with the conformational changes in the S, ultimately leads to the near-complete inactivation of the virus at pH below 3.

Unfortunately, the number of viral particles available for imaging after pH treatment and neutralization was low and did not allow for meaningful S quantification. However, we now comment on the potential S loss in lines 203 - line 207 and thank Reviewer 1 for pointing this out.

Reviewer #2 (Remarks to the Author):

1. Brief Summary

This manuscript investigates the mechanism by which acidic pH inactivates SARS CoV 2. The authors demonstrate that exposure to pH values below 3 rapidly reduces viral infectivity—by 1,000 to 10,000 fold—primarily through inducing partial unfolding of the spike (S) protein, which in turn disrupts its ability to bind to host cells. Notably, the viral envelope and genomic RNA remain intact under these conditions. The study employs a combination of in vitro acid treatment, infectivity assays, cryo-electron tomography, RNase protection assays, and detailed immunofluorescence analyses using multiple S-specific antibodies. A comparison with HCoV 229E (using a Renilla luciferase reporter) underscores a species-specific sensitivity to acidification. In addition, aerosol modeling (via ResAM simulations) is used to explore the potential for environmental acidification as a decontamination strategy.

2. Overall Impression

The manuscript addresses an important and timely question regarding SARS CoV 2 stability and environmental inactivation. The multi-pronged experimental approach is a strength, with complementary techniques supporting the central hypothesis that low pH induces conformational changes in the spike protein leading to loss of receptor binding. However, several aspects of the study—such as the choice of viral isolates, the indirect nature of some measurements, and limitations in the structural data—warrant further clarification and discussion.

3. Specific Comments

1. Control Virus for HCoV 229E:

o The HCoV 229E strain used encodes a Renilla luciferase reporter (HCoV 229E-Ren), while BavPat1 (SARS CoV 2) does not. The manuscript should clarify whether the reporter virus serves as a suitable control for comparing acid inactivation kinetics and whether the reporter might influence viral behavior relative to a wild-type strain.

To control for potential influences of the Renilla gene in our HCoV-229E-Ren strain, we obtained and tested an HCoV-229 wild type strain. The wild type strain was as stable as the HCoV-229E-Ren strain as shown in figure S1C and mentioned in lines 142 – 144.

Figure S1: C) Inactivation curves in FFU/ml of HCoV-229E wild type (HCoV-229E WT), over time at pH 7.1 and 2.0. Data was derived from three independent experiments. Dotted lines indicate the titer measured in the untreated control and the LoD.

2. Choice of SARS CoV 2 Isolate:

o BavPat1 is an early (wild-type) isolate, yet current transmission is predominantly driven by variants like Omicron. The authors should justify the choice of BavPat1 for most experiments and discuss the practical implications of using a wild-type strain versus a currently circulating variant. If feasible, inclusion of additional data using the Omicron strain throughout the study would enhance relevance.

We chose BavPat1 as it grows to higher titers and has a broader range of available commercial antibodies. Further we found that the inactivation kinetic is similar between BavPat1 and the more recent Omicron variant JN.1 (figure S1D-F). However, to address this important point we have now included a set of immunofluorescence experiments confirming that the same structural changes observed with BavPat1 S are also induced in JN.1 S, due to acidic pH (new figure 6A-D mentioned in lines 366 - 381).

Figure 6: Structural changes of SARS-CoV-2 JN.1 S A-D) SARS-CoV-2 JN1 virions were bound to coverslips and subsequently given a pH pulse for 1 min or left untreated (pH7.0, pH2.0 and untreated). The overall intensity ratio of the neutralizing SARS-CoV-2 JN.1 RBD (A) and the SARS-CoV-2 S2 (C) antibody signal to the corresponding N signal was calculated for each image. Additionally, the Manders correlation coefficient M1 was calculated for the neutralizing SARS-CoV-2 JN.1 RBD (B) and the SARS-CoV-2 S2 (D) antibody signal. Error bars represent SD from the mean. All statistical tests were done with a nonparametric one-way ANOVA (Kruskal-Wallis test, two-sided) in GraphPad Prism with all conditions compared to the pH 7.0 control. Data was considered non-significant (ns) if p values were 0.05 or above and significant for p values below 0.05 ($p < 0.05$ (*), $p < 0.005$ (**), $p < 0.0005$ (***)).

3. Impact of UV Inactivation on Cryo-ET:

o In line 178, the authors mention that UV inactivation was performed prior to cryo-electron tomography. It is important to discuss whether UV treatment could affect virion morphology or subtle structural features, thereby influencing the cryo-ET results.

We added a discussion point concerning this in lines 466 - 469.

4. Cryo-ET Resolution and Virion Concentration:

o The resolution of the tomograms appears insufficient to capture subtle differences in spike conformation, and the low concentration of virions further limits quantitative analysis.

We agree that the resolution is not high enough for capturing structural changes in S and comment on this in lines 204 – 206 and 462 – 464. However, from the cryo-ET images we only conclude on overall virion membrane intactness, and this conclusion is further supported by the data from the RNase protection assay. Our conclusions on structural changes in S are based on the data shown in figures 4-6.

5. Evidence for Attachment-Only Conditions:

o The manuscript states that neutralized SARS CoV 2 samples (treated at pH 7.1 or 2.8) were incubated with VAT cells on ice for 1.5 h to permit attachment while preventing internalization (lines 203–204). It would strengthen the study to provide evidence or references supporting that these incubation conditions reliably block internalization yet allow for efficient attachment.

We added the according reference in line 226.

6. Antibody Profiling versus Direct Imaging:

o The assessment of spike conformational changes is largely based on antibody binding profiles. Notably, antibody testing for HCoV 229E is lacking, and reliance on indirect antibody profiling (versus direct high-resolution imaging or biophysical assays) may limit the conclusiveness of the findings. The authors should discuss the limitations of this approach and,

if possible, consider additional methods to directly validate conformational changes in the spike protein.

To tackle these points, we included antibody testing of HCoV-229E (new figure 6E-H and lines 381 – 389) and added a limited proteolysis liquid chromatography-mass spectrometry (LiP LC-MS) experiment with recombinant S to strengthen our conclusions (lines 245 - 263 and new figure 4). For the new figure 4, please also see our response to point 1 of reviewer 1.

Figure 6: Structural changes of HCoV-229E S. E-H) HCoV-229E-Ren virions were bound to coverslips and subsequently given a pH pulse for 1 min or left untreated (pH7.0, pH2.0 and untreated). The overall intensity ratio of the HCoV-229E S1 (E) and the HCoV-229E S2 (G) antibody signal to the corresponding N signal was calculated for each image. Additionally, the Manders correlation coefficient M1 was calculated for the HCoV-229E S1 (F) and the HCoV-229E S2 (H) antibody signal. Error bars represent SD from the mean. All statistical tests were done with a nonparametric one-way ANOVA (Kruskal-Wallis test, two-sided) in GraphPad Prism with all conditions compared to the pH 7.0 control. Data was considered non-significant (ns) if p values were 0.05 or above and significant for p values below 0.05 (p < 0.05 (*), p < 0.005 (**), p < 0.0005 (***)).

7. Evidence of S-RBD Cleavage at Low pH:

The review raises the question of whether there is any evidence that the receptor binding domain (RBD) is cleaved under low-pH conditions. Clarification is needed on whether the observed loss of native RBD epitopes results solely from unfolding or if cleavage may also contribute.

We have excluded cleavage of the RBD with two additional experiments. First, we obtained an additional RBD antibody that has a preference for linear epitopes. Like all other antibodies with linear epitope preferences, but contrary to neutralizing RBD antibodies, this antibody showed an increased signal when comparing pH 2.0 treated SARS-CoV-2 virions to pH 7.0 treated virions (new figure 5B, H and I; description in lines 320 - 322). This indicates that the RBD is still present on viral particles.

Figure 5: SARS-CoV-2 BavPat1 RBD is affected by acidic pH. **B)** SARS-CoV-2 BavPat1 virions were bound to coverslips and subsequently given a pH pulse for 1 min or left untreated (pH7.0, pH 2.6, pH2.0 and untreated). After fixation and permeabilization, virions were stained for N as a control (cyan) and an RBD antibody (magenta) exhibiting a preference for linear epitopes. Shown are representative images from three independent replicates. Scale bar corresponds to 10 μ m. **H-I)** Quantification of three to four images per pH condition from three independent experiments each. In brief, a background cut-off was defined for each channel and subtracted from the images. Only pixels positive for N were considered in the following analyses to further reduce background and processing artefacts. The overall intensity ratio of the RBD (linear epitope preference) signal to the N signal (H) was calculated for each image. Additionally, the Manders correlation coefficient M1 was calculated using Coloc 2 (ImageJ), measuring the fraction of the N intensity signal associated with the the RBD (linear epitope preference) (I) antibody signal. Error bars represent SD from the mean. All statistical tests were done with a nonparametric one-way ANOVA (Kruskal-Wallis test, two-sided) in GraphPad Prism with all conditions compared to the pH 7.0 control. Data was considered non-significant (ns) if p values were 0.05 or above and significant for p values below 0.05 ($p < 0.05$ (*), $p < 0.005$ (**), $p < 0.0005$ (***)).

Second, we analyzed pH treated virions by Western Blot with an S1 antibody (see below). If cleavage occurred within the S1 subunit we would expect to see a band shift in WB. However, this was not the case. To make the manuscript more concise, we decided to only include the new RBD antibody data.

8. Figure Discrepancy:

o In Figure 1c (lower panel), there is an inconsistency where the description indicates a pH of 2.2, but the figure appears to label it as 2.1. The authors should verify and correct this discrepancy to ensure clarity.

The figure caption is correct. We corrected the figure accordingly.

9. General In Vitro Limitations:

o The experiments are conducted under controlled in vitro conditions that may not fully recapitulate the complexity of natural respiratory aerosols (e.g., presence of mucus, varying protein concentrations, and other environmental factors). A discussion of these limitations and their implications for translating the findings into real-world scenarios is warranted.

We added a paragraph addressing this point in lines 484-489.

10. Virus Stock Preparation Variability:

o Variability in virus concentration and stock preparation methods is noted. The authors should clearly describe how these factors were controlled across experiments and discuss their potential impact on the observed inactivation kinetics.

Due to biosafety restrictions, we are unable to purify SARS-CoV-2 stocks by ultracentrifugation. However, when comparing the BavPat1 and JN.1 variant we diluted the BavPat1 stock to match the approximately 1.5 log₁₀ lower titer of JN.1. The dilution did not influence the inactivation kinetic of the BavPat1 strain (figure S1E). Furthermore, we subjected HCoV-229E to different stock dilutions before the pH treatment, again seeing no effect (figure S1G and S1I). Similarly, different concentration methods did not influence the pH stability of HCoV-229E (figure S1H and S1J). We conclude that neither reaction titer nor concentration method influence pH stability (lines 153 – 161). These data were already included in the previous version and are therefore not shown here.

11. Practical Implications for Decontamination:

o Although the study shows that acidic pH effectively inactivates SARS CoV 2 in vitro, achieving such conditions safely in real-world environments poses significant challenges. A more detailed discussion on the feasibility, safety, and limitations of applying these findings to decontamination strategies is recommended.

We added a section addressing this point in lines 430-435.

Reviewer #3 (Remarks to the Author):

In this manuscript, Glas et al. investigate the low stability of SARS-CoV-2 under acidic conditions and its implications for virus inactivation, particularly in the context of airborne transmission. The authors demonstrate that low pH has minimal impact on the integrity of the viral membrane but induces conformational changes in the spike (S) protein, which leads to the inactivation of SARS-CoV-2. This is supported by cryo-electron tomography and various other techniques, providing a relatively comprehensive analysis. Notably, the authors find that SARS-CoV-2 is sensitive to low pH, whereas another human coronavirus, 229E, is not. These findings are significant, but the manuscript would benefit from testing a broader range of variants, particularly currently circulating strains, and exploring practical applications for decontamination strategies. Several issues need to be addressed.

1. Line 125: According to the Methods section, the Omicron variant used in this manuscript is JN.1. Therefore, "Omicron" should be replaced with "JN.1."

2. Why are Omicron variants more transmissible, considering they are more sensitive to acidic pH, even at pH 4.3 (Fig S1E)?

Omicron was replaced by SARS-CoV-2 JN.1 throughout the manuscript.

JN.1, like BavPat1, is insensitive to pH 4.3 since there is no difference in titer after 10 min compared to the pH 7.1 control (figure S1F and line 146-153). While JN.1 is more sensitive at pH 2.8, this pH is not reached in aerosol particles in typical indoor air. Therefore, we do not expect this difference to play a role in typical indoor settings.

3. According to the cryo-EM structure of the S protein of 229E, the S proteins of 229E adopt a closed conformation, while the S proteins of the original SARS-CoV-2 strain likely exist in a mixed open and closed conformation. It would be interesting to select a SARS-CoV-2 variant with a closed conformation and analyze its sensitivity to low pH.

As Omicron variants such as JN.1 are known to preferentially adopt the closed conformation, we believe that open and closed conformation are well reflected by the BavPat1 and JN.1 variant respectively. We added a sentence concerning this in lines 376 – 378.

4. In Fig. 2J, the morphology of the S proteins in the right panel appears to differ from those in the left panel. Additionally, low pH treatment clearly caused conformational changes in the S protein, as shown in Figures 3 and 4. This seems contradictory to the statement in line 194: "no major changes in overall morphology occurred."

We thank the reviewer for highlighting this statement as contradictory. As mentioned in response to point 4 of reviewer 2 we only conclude on virion membrane intactness from the cryo-ET images. The conclusions on structural changes in S are based on the data from figures 4-6. The highlighted sentence was not clear. We have changed the text in lines 212-214 to say "we conclude that the virion membrane of SARS-CoV-2 remains intact upon pH 2 treatment."

5. In lines 198, 219, and 280, "virion integrity" should be revised to "virion membrane integrity" The changes were implemented as suggested.

6. The authors may consider measuring ACE2 binding by various pH-treated viruses using an ELISA.

Although this would be an interesting approach to test S functionality we decided to rather focus on S conformation of the SARS-CoV-2 BavPat1 and JN.1 variants (new figures 4 and 6, please see the new figures above).

7. The effect of pH on S1 shedding should be assessed by pelleting the virions through ultracentrifugation.

Although the proposed experiment would be of great interest to our study, we are not able to ultracentrifuge SARS-CoV-2 due to biosafety restrictions. Instead, we obtained an additional RBD antibody that has a preference for linear epitopes to tackle this question. Like all other antibodies with linear epitope preferences, but contrary to neutralizing RBD antibodies, this antibody showed an increased signal when comparing pH 2.0 treated SARS-CoV-2 virions to pH 7.0 treated virions (figure 5B, H and I). The increased RBD signal for this specific antibody further strengthens our conclusion that the S1 subunit is not shed efficiently from the viral

particles. However, we still cannot fully exclude the possibility and state that in lines 469 – 474. Please see the new figure panels included also in response to comment 7 of reviewer 2.

8. Limited trypsin digestion could be considered to evaluate conformational changes in the S protein induced by low pH treatment.

We thank Reviewer 3 for this idea and highly appreciate it. We implemented this suggestion by subjecting recombinant trimeric S to pH treatment and subsequent limited proteolysis liquid chromatography-mass spectrometry (LiP LC-MS) to evaluate conformational changes. We were able to confirm conformational changes in the S2 and NTD subunit, shown in new figure 4 and lines 245 - 263. Please see the new figure included also in response to point 1 of reviewer 1.

9. In line 377, the manuscript states that BavPat1 and IMV15 were propagated in VAT and Vero E6 cells. The sequences of the SARS-CoV-2 S proteins should be verified by sequencing to rule out the possibility of partial furin site deletions.

Sequencing of the used BavPat1 and JN.1 (IMV15) stocks revealed that BavPat1 had retained the furin cleavage site, while our JN.1 isolate had lost the furin cleavage site already in the isolation process. We now mention this fact in the methods section (line 536 – 538). Given that BavPat1 and JN.1 display similar inactivation kinetics and the same partial unfolding of spike at pH 2 we speculate that the presence or absence of the furin cleavage site does not impact this feature of the virus.

10. In Fig. 4I, three preparations of SARS-CoV-2 show significant differences in the S1/S ratio. Additional western blot analysis using an anti-S2 antibody might be considered.

As suggested, we repeated the WB analysis with an S2 antibody (figure 5L and M). Since the signal from the S2 antibody was stronger than from the original S1 antibody (which was close to the detection limit), we concluded that the quantification with the help of the S2 antibody is more robust. We therefore replaced the original data with a new blot using the S2 antibody.

Figure 5: L) WBs of 3 different SARS-CoV-2 stock preparations (1, 2 and 3) showing N and S. Full-length S as well as the S2 subunit were detected with an S2-specific antibody. M) Full lengths S and S2 WBs were quantified using Image Studio Lite and the S2 to full-length S ratio was calculated. Error bars represent SD from the mean.